# A time-dependent diffusion MRI signature of axon caliber variations and beading

Hong-Hsi Lee [1✉], Antonios Papaioannou[1], Sung-Lyoung Kim[1], Dmitry S. Novikov [1] & Els Fieremans[1]

MRI provides a unique non-invasive window into the brain, yet is limited to millimeter resolution, orders of magnitude coarser than cell dimensions. Here, we show that diffusion MRI is sensitive to the micrometer-scale variations in axon caliber or pathological beading, by identifying a signature power-law diffusion time-dependence of the along-fiber diffusion coefficient. We observe this signature in human brain white matter and identify its origins by Monte Carlo simulations in realistic substrates from 3-dimensional electron microscopy of mouse corpus callosum. Simulations reveal that the time-dependence originates from axon caliber variation, rather than from mitochondria or axonal undulations. We report a decreased amplitude of time-dependence in multiple sclerosis lesions, illustrating the potential sensitivity of our method to axonal beading in a plethora of neurodegenerative disorders. This specificity to microstructure offers an exciting possibility of bridging across scales to image cellular-level pathology with a clinically feasible MRI technique.

[1] Center for Biomedical Imaging and Center for Advanced Imaging Innovation and Research (CAI2R), Department of Radiology, New York University School of Medicine, New York, NY 10016, USA. ✉email: Honghsi.Lee@nyulangone.org

Diffusion magnetic resonance imaging (dMRI) is sensitive to the micrometer length scale via the commensurate diffusion length and, as such, is a promising in vivo technique for evaluating micrometer-scale structural features (the so-called tissue microstructure) of biological tissues in health and disease. The sensitivity to tissue microstructure, however, is indirect, due to averaging of the local diffusion propagator over the millimeter-sized MRI imaging voxel. Biophysical modeling of the diffusion signal in biological tissue[1–4] is therefore essential for quantification of cellular parameters and to gain specificity to cellular changes in development, aging, and pathology. This raises the critical question of which salient features of cells or tissues can be robustly retrieved across the gap of three orders of magnitude in spatial scales and what the essential assumptions are to construct the most parsimonious biophysical models, thereby attaining the highest precision without losing accuracy.

Axonal microgeometry in brain white matter (WM) is special, as axonal diameters are much thinner than the clinically attainable diffusion length $L_d(t) \sim 10$ μm. Hence, intra-axonal diffusion has been described[5] as occurring within infinitely narrow featureless impermeable tubes—dubbed "sticks"—inside which diffusion is effectively one-dimensional and Gaussian, completely determined by a constant diffusion coefficient. This simplified viewpoint—a cornerstone ingredient of the so-called WM Standard Model[4]—has been the basis for WM dMRI modeling over more than a decade, approximating the net intra-axonal space (IAS) within an MRI voxel as a collection of these sticks. In this picture, sticks are deemed non-exchanging with extra-axonal water and their overall orientation is modeled either by a specific distribution function, such as the Watson distribution[6], or by using spherical harmonics[7–11]. The stick model parameters, such as the the intra-stick diffusion coefficient and the orientation dispersion, provide biophysical significance, as they make dMRI specific to axonal pathology.

Although suggested by N-acetyl-L-aspartate (NAA) experiments 16 years ago[5], for water dMRI the stick picture has been validated only recently. Such validation is challenging, as fit quality alone is insufficient to validate a model. Selecting models becomes feasible by testing their unique functional forms in the domain where the dependence on experimental parameters clearly reveals their assumptions[12]. Borrowing this methodology from the physical sciences, the assumptions of the existence of sticks (i.e., of the locally one-dimensional (1d) water diffusion) and of negligible exchange between sticks and extra-axonal water on the time scale of clinical dMRI have been validated in vivo in human brain WM by observing the $1/\sqrt{b}$ dMRI signal scaling (ideal stick response) at very strong diffusion weighting $b \sim 10$ ms/μm$^2$ [11,13].

How adequate is the picture of featureless sticks? In this work, we show that the diffusion inside the IAS along axons is non-Gaussian at clinically employed diffusion times $t \sim 10$–$100$ ms and identify the dominant geometric features for this non-Gaussianity, which can thus be quantified with a dMRI measurement. For that, we focus on varying the diffusion time $t$ rather than on increasing the dMRI wave vector $q$.

The absence of time dependence in the overall diffusivity $D$ would signify Gaussian diffusion in every tissue compartment, whereas the presence of $t$-dependence would reveal microscopic heterogeneity being coarse-grained by diffusion in at least one of the compartments[3,4,14,15]. So far, many WM studies focused on the diffusion time dependence perpendicular to axons, to probe the inner axon diameter[16–19] and the packing correlation length of the extra-axonal space[19–22]. Recently, however, the diffusion tensor eigenvalue parallel to major human WM tracts was found to decrease by 10–15% over the range $t = 50$–$600$ ms using

stimulated-echo (STE) dMRI[21]. This nontrivial time dependence along the tract could not be explained solely by the fiber dispersion, i.e., by the locally transverse $t$-dependent contributions projected onto the tract direction. Rather, the observed non-Gaussian diffusion along axons suggests that either the extra-axonal space or the IAS (the sticks) should be augmented to incorporate micrometer-scale restrictions along the axon bundle direction.

What are these restrictions? According to the effective medium theory of ref. [15], observing a specific power-law time dependence of the 1d diffusivity (along fiber),

$$D(t) \simeq D_\infty + c \cdot t^{-\vartheta}, \quad \vartheta = \frac{1}{2} \qquad (1)$$

approaching its long time limit $D_\infty$ with the strength $c$ of restrictions, is a signature of a short-range disorder of the placement of the restrictions. However, this general theory does not reveal the exact source of these restrictions, be it the mitochondria or beads, or axonal undulations, or the disordered extra-axonal space geometry.

Here we show that the IAS diffusion $t$-dependence has the form (1) and is most sensitive to axon caliber variations, a vital signature of normal axonal microgeometry, which may be altered in pathology, turning into axonal beading. The strength $c$ of restrictions to axial diffusion emerges from randomly placed local axon caliber maxima and depends on caliber variation. To verify the power law (1) and attribute it to axon caliber variation, we evaluate the effect of axon shape on diffusion by developing, for the first time to our knowledge, a full three-dimensional (3d) Monte Carlo (MC) simulations of dMRI in a realistic microgeometry based on 3d electron microscopy (EM) segmentation[23] of mouse brain corpus callosum (CC).

This study is organized as follows. First, we link the power-law dynamics of the time-dependent diffusivity along axons, $D(t)$, with the power spectrum of restrictions to 1d diffusion, allowing us to predict the time dependence from the EM-derived axonal structure. Further, we perform MC simulations in realistic 3d IAS to calculate dMRI-related metrics, such as $D(t)$ and time-dependent kurtosis, $K(t)$, and study their unique functional forms. To better understand the origin of the diffusion time dependence along axons, we separately evaluate the effect of mitochondria, caliber variation, undulation, and axonal orientation dispersion. Our simulations reveal axon caliber variation as the dominant source of time-dependent diffusion along axons. Finally, we show that theory and simulations are consistent with in vivo brain data in 15 healthy subjects acquired using pulsed-gradient spin-echo (PGSE) dMRI at clinical diffusion times. The change of diffusivity time dependence due to specific pathology is also demonstrated in pilot data of five multiple sclerosis (MS) patients. In conclusion, we combine our theory, MC simulations, and clinical dMRI measurements into an overarching picture of a fundamental biophysical phenomenon—axonal caliber variation manifested by a signature power-law exponent $\vartheta = 1/2$—providing a remarkable specificity of a macroscopic dMRI measurement to a particular geometric feature of micrometer-scale axonal microstructure.

## Results

**From axonal structure to the diffusive dynamics.** The power-law tail of the diffusion time dependence, Eq. (1), is determined by the *structural universality class* of the medium[15], with dynamical exponent

$$\vartheta = \frac{p + d}{2} \qquad (2)$$

in $d$ spatial dimensions. It was noted that randomly looking media can be random in a few distinct ways and, thereby, can be classified into a few so-called universality classes (analogously to the universality classes in the theory of critical phenomena). A structural universality class is defined by the structural exponent $p$, describing the statistics of long-range structural fluctuations. Technically, $p$ is defined via the asymptotic behavior[15]

$$\Gamma(\mathbf{k}) = \int d^d\mathbf{x}\ \Gamma(\mathbf{x})\ e^{-i\mathbf{k}\mathbf{x}} \sim k^p, \quad k \to 0$$

of the *power spectrum* $\Gamma(k)$ of the medium at low wave vector $k$—equivalently, the asymptotic behavior of the density–density correlation function

$$\Gamma(\mathbf{x}) = \langle \rho(\mathbf{x}_0 + \mathbf{x})\rho(\mathbf{x}) \rangle_{\mathbf{x}_0} \tag{3}$$

at large distances $|\mathbf{x}|$ (here, the average $\langle \dots \rangle$ is performed over the initial point $\mathbf{x}_0$). Molecular displacement over the diffusion length $L_d(t)$ probes the distances $|\mathbf{x}| \sim L_d(t)$ and thereby samples the statistics of spatial density fluctuations. Thus, Eq. (2) provides the fundamental connection between structure and dynamics.

To determine the structural universality class of the microgeometry along axons, we begin from the $d = 3$ density–density correlation function, Eq. (3), where $\rho(\mathbf{x})$ is the 3d binary mask of an axially symmetric cylinder with radius variation $r(z)$ constructed from realistic axons along axonal axis $z$. We would like to construct the corresponding $d = 1$ power spectrum

$$\Gamma_{1d}(k_z) = \frac{1}{\overline{A}} \int \Gamma_{3d}(\mathbf{x})\ e^{-ik_z z} d^3\mathbf{x} = \frac{|\rho(\mathbf{k}_\perp, k_z)|^2}{V \cdot \overline{A}}\bigg|_{\mathbf{k}_\perp = 0} \tag{4}$$

relevant at long distances $\sim 1/k_z$ exceeding the transverse dimensions of axons, when the diffusion becomes effectively one-dimensional. Hence, in Eq. (4), $\mathbf{k}_\perp = (k_x, k_y)$ is set to 0 as the diffusive motion is fully coarse-grained within the axonal cross-section on time scales much faster than the relevant diffusion times. In this equation, we also used the Wiener–Khinchin theorem $\Gamma(\mathbf{k}) = |\rho(\mathbf{k})|^2/V$, where $V$ is the (axonal) volume. Finally, as our resulting object $\Gamma_{1d}(k_z)$ is a 1d power spectrum, it should have dimensions of length; hence, we normalize by the mean cross-sectional area (CSA) $\overline{A}$.

The restrictions in general can be provided by any kind of microstructural inhomogeneity. Here, they are interpreted as coming from focal swellings or beads (caliber maxima) and constrictions (minima) along axons. Below we study the behavior

$$\Gamma_{1d}(k_z)\big|_{k_z \to 0} \sim k_z^p, \quad k_z \to 0 \tag{5}$$

which will determine the structural exponent $p$ determining the universality class of the $d = 1$ microgeometry.

**Axonal structure analysis reveals short-range disorder.** To estimate the structural exponent $p$ in Eq. (5), we calculate the power spectrum using the radius variation along 227 segmented myelinated axons aligned with the $z$-axis (Fig. 1a, b)[23]. Practically, each axon's inner radius variation $r(z)$ (Fig. 1c) is first scaled by a factor based on the ratio of each axon's volume to the mean volume, ensuring that every axon has the same volume after normalization (Fig. 1d). Next, the normalized radius variations are randomly concatenated along the $z$-axis. Finally, the concatenated normalized radius variation is rotated around the $z$-axis to generate an axially symmetric 3d binary mask $\rho(\mathbf{r})$. The 1d power spectrum $\Gamma_{1d}(k_z)$ is calculated according to Eq. (4).

The power spectrum $\Gamma_{1d}(k_z)$ (Eq. (5)) along the concatenated axon with normalized radii approaches a plateau at low $k_z$ (Fig. 1f) and indicates a structural exponent

$$p = 0. \tag{6}$$

Equation (2) thus yields dynamical exponent $\vartheta = 1/2$ in dimension $d = 1$. *Our prediction* of $\vartheta = 1/2$ and of the power-law tail in Eq. (1) will be tested below using MC simulations and dMRI measurements in human subjects.

The low-$k_z$ plateau demonstrates that restrictions along axons are randomly distributed with a finite correlation length, which is by definition a short-range disorder class of randomness. The level of the plateau is determined by the mean $\bar{a}$ and the variance $\sigma_a^2$ of the distance between restrictions (see Eq. [S13] and following derivations in the Supplementary Information of ref. [15]), as well as by the average restriction width $\bar{l}$ (see Eq. (47) in Appendix B of ref. [20] with restriction "shape" $\nu(k)\big|_{k \to 0} \to \bar{l}$):

$$\Gamma_{1d}\big|_{k_z \to 0, p=0} \simeq \frac{\sigma_a^2}{\bar{a}^2} \cdot \frac{\bar{l}^2}{\bar{a}}. \tag{7}$$

The normalized power spectrum in Fig. 1f has a low-$k_z$ plateau at $\Gamma_{1d}(k_z)/\bar{a} \simeq 0.25$. We can further calculate the corresponding average restriction width $\bar{l} \simeq 5.6\ \mu m$, given that $\bar{a} \simeq 5.70\ \mu m$ and $\sigma_a \simeq 2.88\ \mu m$ (Fig. 1e) are estimated by locating the local maxima of axon caliber variations. In addition, the power spectrum $\Gamma_{pos}(k_z)$ of positions of the local maxima in concatenated caliber variations (normalized as in Fig. 1 and Eq. [S11] of ref. [15]) also shows a low-$k_z$ plateau at $\Gamma_{pos}\big|_{k_z \to 0, p=0}$. $\bar{a} \simeq \sigma_a^2/\bar{a}^2$, indicating a structural exponent $p = 0$ along axons.

**Simulations validate time-dependent diffusion due to caliber variation.** Numerical simulations for validating dMRI in brain microstructure (reviewed by ref. [24]) have been performed either in two-dimensional (2d) or 3d simple geometries, or in combinations thereof. In particular, the axonal shape is typically modeled by artificial geometries. Recently, benefiting from the advances in microscopy, MC simulations were performed in 2d realistic microgeometry of neural tissue reconstructed from light microscopy[25] or EM[26], and also in 3d realistic microstructure of astrocytes reconstructed from confocal microscopy[27]. However, the crucial piece of the validation puzzle—simulations in 3d realistic EM-based neuronal tissue microstructure (e.g., Fig. 1a, b)—have been missing so far.

In the "Methods", MC simulations in realistic microstructure, we describe our IAS segmentation and the MC simulations algorithm.

To explore the possible cause of diffusion time dependence along axons, we compare simulation results of four different microgeometries (Fig. 2a):

I. Original IAS segmentation from EM, with transverse relaxation time $T_{2a} = 80$ ms and intrinsic diffusivity $D_a = 2\ \mu m^2/ms$ in the cytoplasm[28], $T_{2m} = 20$ ms and $D_m = 0.13\ \mu m^2/ms$ in the mitochondria, and fully permeable mitochondrial membrane[29]. This segmentation is the closest to reality and serves as the main result.

II. The same IAS, but with no $T_2$ contrast and intrinsic diffusivity difference between mitochondria and axoplasm ($T_{2a} = T_{2m} = 80$ ms, $D_a = D_m = 2\ \mu m^2/ms$).

III. Axially symmetric IAS with the same caliber variation (i.e., the same $z$-dependent CSA) as in the original IAS, but no undulation.

IV. IAS includes undulation and preserves volume, but has no caliber variation. The axonal skeleton describing the undulation is constructed by connecting the center of mass of each cross-section and smoothed along the axon by a Gaussian filter of a standard deviation $\sigma = 1\ \mu m$.

All fibers are aligned to the $z$-axis and the orientation dispersion is not considered when comparing the above four

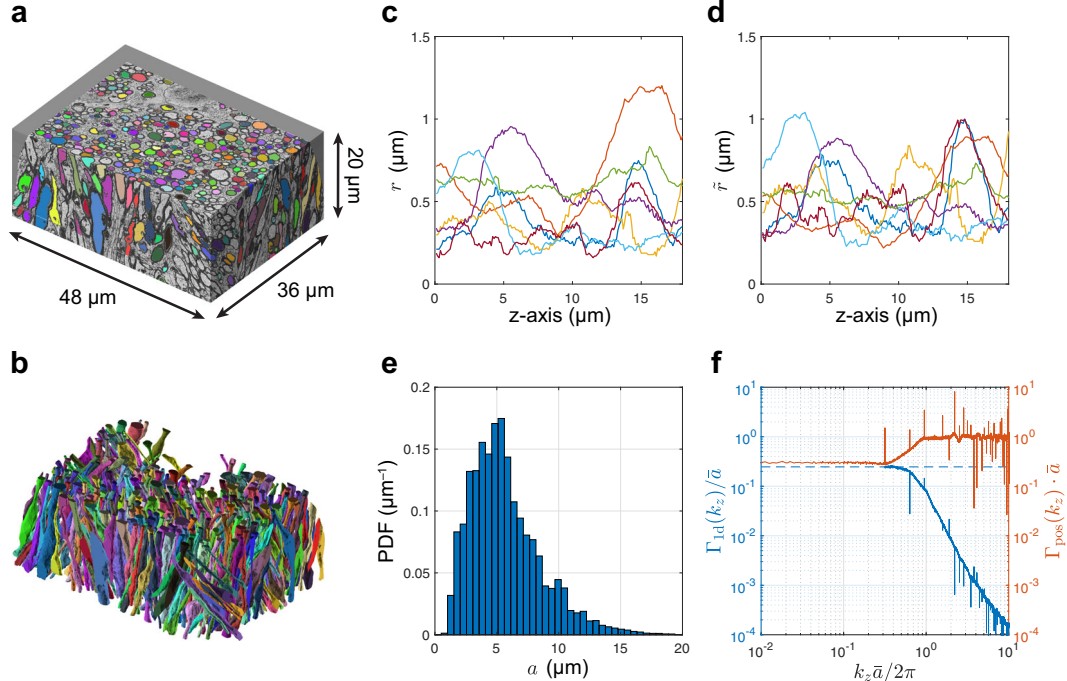

**Fig. 1 Structural analysis of axons segmented from female mouse brain corpus callosum EM reveals that the 1d placement of caliber variations exhibits short-range disorder, characterized by a finite correlation length. a** 3d EM image with segmented axons passing through the central slice. **b** 3d representation of the intra-axonal space (IAS) segmentation yielding 227 axons that are long enough to pass through all slices (>20 μm). **c** Radius variation $r(z)$ and **d** normalized radius variation $\tilde{r}(z)$ along seven selected axons. **e** Histogram of distances $a$ between local radius maxima along all 227 segmented axons. **f** The power spectrum $\Gamma_{1d}(k_z)$ along all axons (blue curve, truncated at $k_z \sim 1/L$ due to limited individual axon's length $L$) shows a plateau at low $k_z \ll 1/\bar{a}$ with $\bar{a}$ the mean distance between restrictions (blue dashed line), indicating a structural exponent $p = 0$, which corresponds to the short-range disorder and leads to the dynamical exponent $\vartheta = 1/2$ in Eq. (1), cf. Eq. (2). Serving as a reference, the red curve shows the power spectrum $\Gamma_{pos}(k_z)$ of the restriction positions[15] with a plateau at low $k_z$ as well. Panels **a** and **b** are adapted from ref. [23] by permission from Springer Nature: Brain Structure and Function, Copyright, 2019.

cases. The effect of dispersion is considered separately in Fig. 2e–h.

In the microstructure based on realistic IAS in Fig. 2a (I–IV), the simulated overall $D(t)$ (from all axons) exhibits a notable time dependence, which scales as $1/\sqrt{t}$ (Fig. 2b). This is in agreement with our theoretical prediction of Eq. (1), corresponding to the dynamical exponent $\vartheta = 1/2$ and the structural exponent of Eq. (6), and confirms our expectations that the restrictions to diffusion along axons are due to short-range disorder. The corresponding bulk diffusivity $D_\infty$ and strength $c$ of restrictions (Eq. (1)) for all axons are listed in Table 1, based on Eq. (11), individual axon's volume fraction $f_i$, and parameters $(D_{i,\infty}, c_i)$ obtained by fitting Eq. (1) to individual axon's $D_i(t)$.

The simulated $D(t)$ with or without considering low $T_2$ and low intrinsic diffusivity in the mitochondria (I and II) shows similar diffusivity values and time dependence (Fig. 2b and Table 1). Similarly, compared with structure I, $D(t)$ of axially symmetric cylinders with only caliber variation (III) has slightly larger diffusivity values and very similar time dependence. On the other hand, $D(t)$ of undulating fibers with no caliber variation (IV) shows much larger diffusivity values and negligible time dependence (~0.05% diffusivity change at $t = 20$–100 ms), indicating that caliber variation is the main cause for the observed time dependence. For the microgeometry I in Fig. 2a, the radius variation along individual axon, i.e., coefficient of variation of radii $CV(r)$, highly correlates with the relative diffusivity variation, i.e., $\zeta \equiv (D_0 - D_{i,\infty})/D_{i,\infty}$ with the intrinsic diffusivity $D_0$, via a quadratic function (Pearson's $R = 0.8917$ for $\zeta$ and $CV^2(r)$ in Fig. 2d), a relation derived in Eq. (15) in "Methods." In the microgeometry I, $D_0$ is approximated by the volume-weighted sum of intrinsic diffusivities in IAS and mitochondria: $D_0 \simeq (1 - f_m)D_a + f_m D_m \simeq 1.89 \pm 0.06\ \mu m^2/ms$, with the mitochondrial to IAS volume ratio $f_m \simeq 6\%$ reported in Supplementary Fig. 1c.

It is essential to evaluate the *effect of fiber orientation dispersion*, because the diffusion time dependence transverse to individual axons could be projected to the main direction of the whole fiber bundle, confounding the $1/\sqrt{t}$ dependence in Eq. (1). To evaluate this effect (Fig. 2e–h), segmented axons in Fig. 2a, scenario I, were oriented based on a Watson distribution with concentration parameters $\kappa = [\infty, 15.4, 4.7, 1.65]$ for cases of no dispersion up to high dispersion, corresponding to the overall polar dispersion angles $\theta = [0°, 15°, 30°, 45°]$, defined by $\theta \equiv \cos^{-1}\sqrt{\langle \cos^2\theta \rangle}$[10,23]. As a reference, the dispersion angle in the mouse brain CC[23] is ~24°, corresponding to $\kappa \sim 6.9$. This preserves the $D(t)$ scaling as $1/\sqrt{t}$, which overall decreases with increasing dispersion angle (Fig. 2e), as manifested by the corresponding fit parameters in Eq. (1) (Table 2), bulk diffusivity in long time limit and strength of restrictions: $D_\infty$ and $c \propto \langle \cos^2\theta \rangle$ (Eq. (12) and Fig. 2g, h). In particular, the estimate of $c$ slightly deviates from this relation (Fig. 2h) due to an extra $1/t$ term contributed by the diffusion transverse to individual axons, especially for the high dispersion case (large $\theta$, small $\langle \cos^2\theta \rangle$). Accounting for this small effect by using Eq. (13), the corrected value of $c$ restores the relation.

The finite value of the time dependence amplitude $c$ in Eq. (1) corresponds to about 4.4% $D(t)$ change over $t = 20$–100 ms time range. In particular, the axial diffusivity change $\propto \Delta(1/\sqrt{t}) \sim \Delta t \cdot t^{-3/2}$ is even larger at short diffusion times. Including time

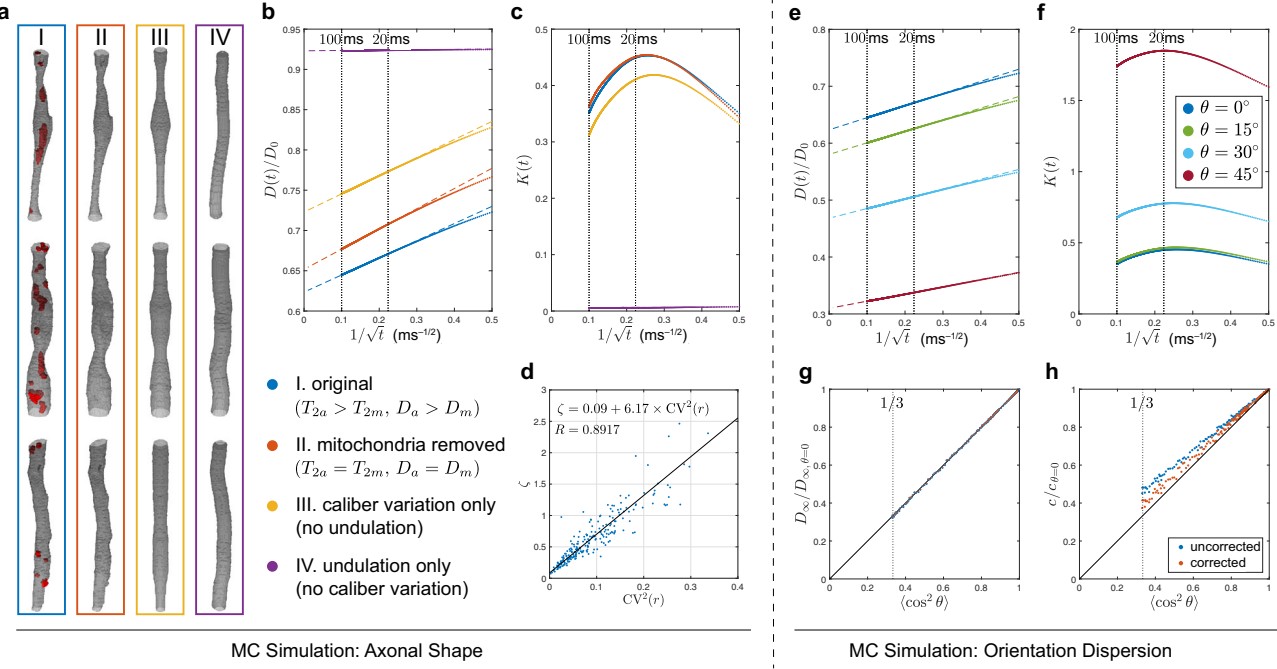

**Fig. 2 MC simulations inside 227 axons to study time-dependent diffusion along axons.** *Which microstructural feature of the axonal shape explains the observed diffusion t-dependence?* **a** Starting from the geometry of axons segmented from EM, four different types of microgeometries were created as follows (see text): (I) with or (II) without considering shorter transverse relaxation time $T_{2m}$ and smaller intrinsic diffusivity $D_m$ in mitochondria (red), and derived synthetic axons with (III) only caliber variation or (IV) only axonal undulation. **b** The simulated $D(t)$ along axons for scenarios (I–IV) plotted as $1/\sqrt{t}$. The linear scaling at long $t$ points to 1d short-range disorder along axons. Dashed lines are the asymptotes based on Eq. (1) and Eq. (11), with fit parameters $D_\infty$ and $c$ in Table 1. Remarkably, $D(t)$ is mostly influenced by caliber variation, as it becomes much weaker in scenario IV when caliber variations are removed. **c** The simulated $K(t)$ along axons for scenarios (I–IV) plotted as $1/\sqrt{t}$, showing a non-monotonic change with $t$. Although removing undulations (III) slightly lowers the kurtosis, its significant reduction and altering the $t$-dependence occurs when caliber variations are removed (IV). Both **b** and **c** indicate that axon caliber variations are the dominant contributions to the IAS time dependence. **d** Illustration of the relation $\zeta \propto \mathrm{CV}^2(r)$ in the original IAS (scenario I), see Eq. (15) derived in "Methods" using coarse-graining arguments. *What is the effect of orientation dispersion?* **e** Axon bundles with axially symmetric orientation dispersion were created with orientation distributions of polar angles $\theta = [0°, 15°, 30°, 45°]$. The simulated $D(t)$ along axons scales as $1/\sqrt{t}$ and decreases with the dispersion angle. The dashed lines are predictions based on Eq. (1) and Eq. (11), with parameters $D_\infty$ and $c$ shown in Table 2. **f** The simulated $K(t)$ along axons increases with the dispersion angle. **g** The bulk diffusivity for $t \to \infty$, $D_\infty \propto \langle \cos^2 \theta \rangle$, Eq. (12). **h** The strength of restrictions, $c$, slightly deviates from this proportionality relation in Eq. (12) (blue). Accounting for the higher-order $1/t$ term in Eq. (13), the corrected value of $c$ restores Eq. (12) (red).

**Table 1 Fit parameters of the time-dependent axial diffusivity $D(t)$ in our simulations in four axonal microgeometries (I–IV) specified in Fig. 2a, b.**

| Microgeometry (Fig. 2a) | $D_\infty$ (μm²/ms) | $c$ (μm² · ms⁻¹/²) |
|---|---|---|
| I. $T_{2a} > T_{2m}$, $D_a > D_m$ | 1.25 | 0.426 |
| II. $T_{2a} = T_{2m}$, $D_a = D_m$ | 1.30 | 0.502 |
| III. Caliber variation only | 1.45 | 0.450 |
| IV. Undulation only | 1.85 | 0.009 |

**Table 2 Fit parameters of the time-dependent axial diffusivity $D(t)$ in our simulations in 3d (Fig. 2e).**

| Dispersion angle $\theta$ (°) | $D_\infty$ (μm²/ms) | $c$ (μm² · ms⁻¹/²) |
|---|---|---|
| 0 | 1.25 | 0.426 |
| 15 | 1.16 | 0.407 |
| 30 | 0.94 | 0.343 |
| 45 | 0.62 | 0.253 |

Simulation results for dispersion angles $\theta = 15°$–$30°$ are consistent with the human brain PGSE data in genu, cf. Table 3.

dependence for the intra-axonal compartment is therefore especially important for animal imaging[30] and for human dMRI[31] at relatively short diffusion times, achievable on high-gradient systems.

For the time dependence of higher-order cumulants of the intra-axonal signal, similar observation are made for the simulated overall kurtosis $K(t)$ in Fig. 2c: in realistic IAS (Fig. 2a, I and II), $K(t)$ has almost the same values and overall $t$-dependence with or without considering low $T_2$ and low intrinsic diffusivity in mitochondria (Fig. 2c). Similarly, compared with microgeometry I, the scenario with no axonal undulation (III)

results in slightly smaller kurtosis values and similar $K(t)$ form. On the other hand, the scenario with no caliber variation (IV) shows much smaller kurtosis values and a totally different $K(t)$ form. These results indicate that the kurtosis time dependence along realistic axons largely depends on caliber variation rather than on axonal undulation, with a small effect of low $T_2$ and low intrinsic diffusivity in mitochondria. For a fiber bundle with orientation dispersion, the simulated overall $K(t)$ increases with the dispersion angle (Fig. 2f), especially for $\theta \gtrsim 30°$.

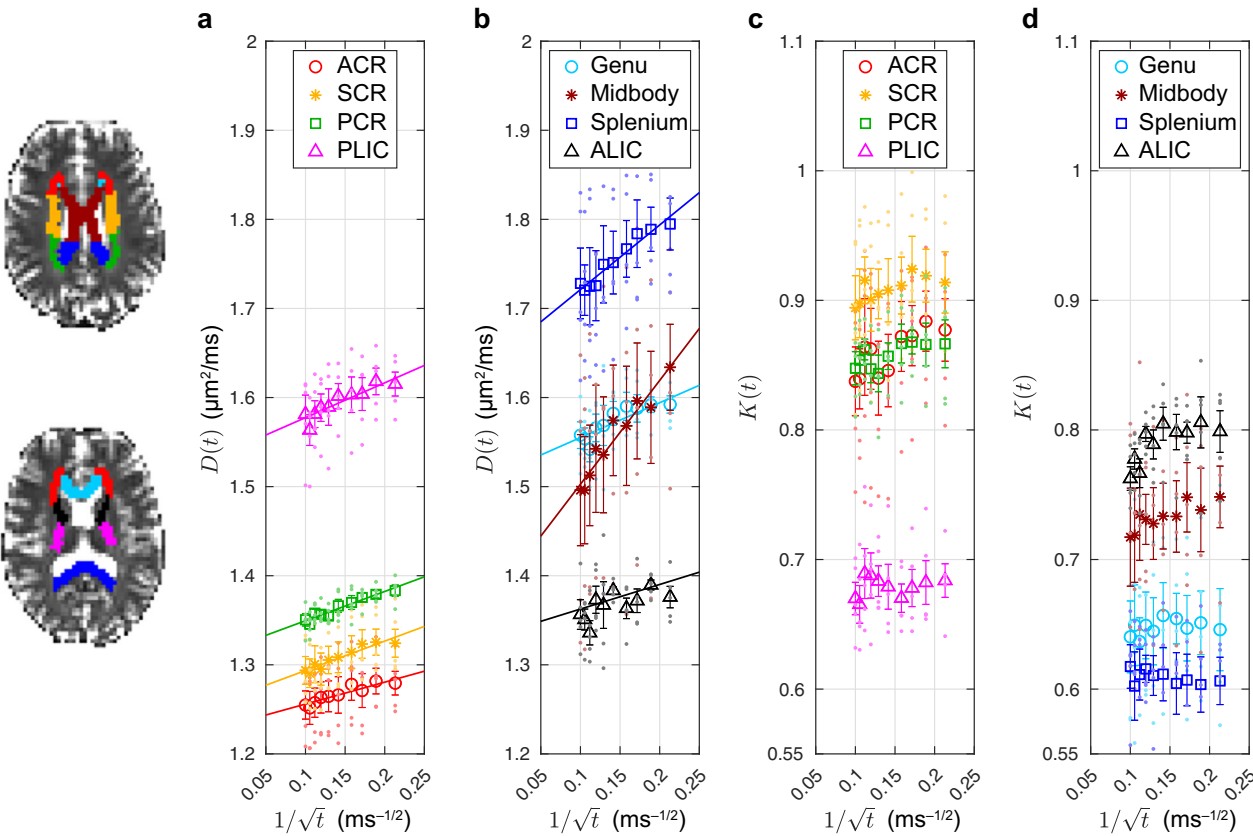

**Fig. 3 Time-dependent axial diffusivity $D(t)$ and axial kurtosis $K(t)$ measured in vivo in brain WM of five healthy subjects using monopolar PGSE.**
**a**, **b** In all WM ROIs, the axial diffusivity scales as $1/\sqrt{t}$ in Eq. (1) (*P*-value < 0.05, Table 3), confirming our prediction that the universality class along WM axons is 1d short-range disorder, cf. Fig. 1. The fit parameters are summarized in Table 3. **c**, **d** In all WM ROIs, the axial kurtosis is ~0.8, demonstrating the non-Gaussian diffusion along axons. Points are plotted representing ROI-values of each subject, along with corresponding mean values (symbol and color in legend) and error bars indicating the SE over five subjects. (ACR/SCR/PCR, anterior/superior/posterior corona radiata; ALIC/PLIC, anterior/posterior limb of the internal capsule; genu/midbody/splenium of CC).

---

**Table 3 Fit parameters of the time-dependent axial diffusivity $D(t)$ in human brain data measured using monopolar PGSE (Fig. 3a, b).**

| ROI | *P*-value | $D_\infty$ (µm²/ms) | $c$ (µm² · ms$^{-1/2}$) |
|---|---|---|---|
| ACR | 6.3e − 5 | 1.231 (0.005) | 0.246 (0.034) |
| SCR | 1.3e − 5 | 1.261 (0.006) | 0.330 (0.038) |
| PCR | 3.3e − 6 | 1.317 (0.005) | 0.329 (0.033) |
| PLIC | 1.2e − 4 | 1.538 (0.010) | 0.390 (0.062) |
| Genu | 4.2e − 4 | 1.516 (0.011) | 0.391 (0.069) |
| Midbody | 5.2e − 6 | 1.386 (0.016) | 1.17 (0.10) |
| Splenium | 1.9e − 6 | 1.649 (0.009) | 0.725 (0.058) |
| ALIC | 1.8e − 2 | 1.335 (0.020) | 0.276 (0.129) |

SEs are shown in the parenthesis.

---

Focusing on the realistic microgeometry I without considering dispersion (dark blue data points in Fig. 2c, f), the simulated overall $K(t)$ (~0.4 at $t = 20$–100 ms) consists of two parts as follows: (1) the inter-compartmental contribution originating from the diffusivity differences between multiple axons (first right-hand-side term in Eq. (10b)) and accounting for 24–37% of $K(t)$ at $t = 20$–100 ms and (2) the intra-compartmental contribution originating from individual axon's axial kurtosis (second right-hand-side term in Eq. (10b)) and accounting for 76–63% of $K(t)$ at $t = 20$–100 ms.

**In vivo MRI demonstrates diffusion time dependence along axons**. The time-dependent axial diffusivity $D(t)$, measured by monopolar PGSE in the human brain WM (Fig. 3a, b), were averaged over five healthy subjects ($n = 5$) and plotted with respect to $1/\sqrt{t}$. In all studied WM regions of interest (ROIs), the axial diffusivity time dependence demonstrates a $1/\sqrt{t}$ power-law relation in Eq. (1) (*P*-value < 0.05, Table 3), indicating that the universality class along human WM axons is short-range disorder (randomly distributed tissue inhomogeneity) in 1d, corresponding to a dynamical exponent $\vartheta = 1/2$. The fit parameters in the different WM ROIs ($D_\infty$, $c$) are shown in Table 3. Figure 3c, d also shows that the axial kurtosis in WM ROIs from the same in vivo measurements is ~0.8 and varies over diffusion time in some ROIs, demonstrating non-Gaussian diffusion along axons. The data of ten additional subjects scanned with higher resolution are in Supplementary Fig. 2.

To further demonstrate the regional variation, Fig. 4 shows the variation across the nine sub-regions of CC (G1/G2/G3 for the genu, B1/B2/B3 for the midbody, and S1/S2/S3 for the splenium) in the time-dependent parameters for each subject. The bulk diffusivity $D_\infty$ in $t \to \infty$ limit has a high–low–high pattern in genu–midbody–splenium in all subjects (Fig. 4b), whereas the strength $c$ of restrictions along axons has a low–high–low pattern in most of the subjects (Fig. 4c).

**Time-dependent diffusion parameters alter in multiple sclerosis**. To evaluate the sensitivity of the time-dependent

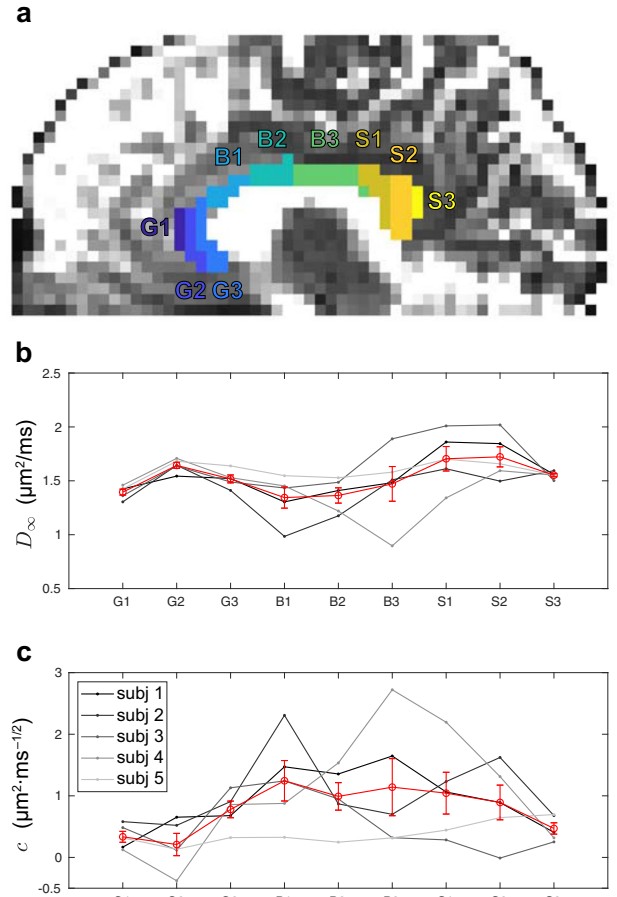

**Fig. 4 Time-dependent parameters, $D_\infty$ and $c$ in Eq. (1), estimated in vivo in brain CC of five healthy subjects using monopolar PGSE. a** Sub-regions of CC, including G1, G2, G3 for the genu, B1, B2, B3 for the midbody, and S1, S2, S3 for the splenium. **b** The bulk diffusivity $D_\infty$ shows a high–low–high pattern in CC. **c** The strength $c$ of restrictions shows a low–high–low pattern in CC. The red data point is the mean of five subjects and the error bar indicates the SE of five subjects.

diffusion parameters to pathology, the time-dependent axial diffusivity $D(t)$ was measured by monopolar PGSE in MS lesions and normal-appearing WM (NAWM) in five MS patients ($n = 5$). $D(t)$ averaged over subjects is plotted with respect to $1/\sqrt{t}$ in Fig. 5a, confirming that both in MS lesions and in NAWM, $D(t)$ obeys the power-law relation in Eq. (1), with $P$-values = 0.042 and 0.012, respectively.

The fit parameters ($D_\infty$, $c$, Fig. 5b, c) estimated individually in MS patients are compared between MS lesions and NAWM. The bulk diffusivity $D_\infty$ along axons in $t \to \infty$ limit is significantly larger in MS lesions than that in NAWM ($P$-value = 0.031, Fig. 5b). Furthermore, the strength $c$ of restrictions along axons is significantly smaller in MS lesions than that in NAWM ($P$-value = 0.031, Fig. 5c).

## Discussion
The time-dependent dMRI signal measured in vivo in brain WM provides a signature for along-axon caliber variation. The specificity to this microstructural feature is determined here from a characteristic power-law decay of the diffusivity and validated by performing realistic MC simulations of diffusion inside axons from 3d EM images of mouse brain. In particular, our simulation results are consistent with in vivo measurements and the corresponding theoretical prediction that diffusion along axons is

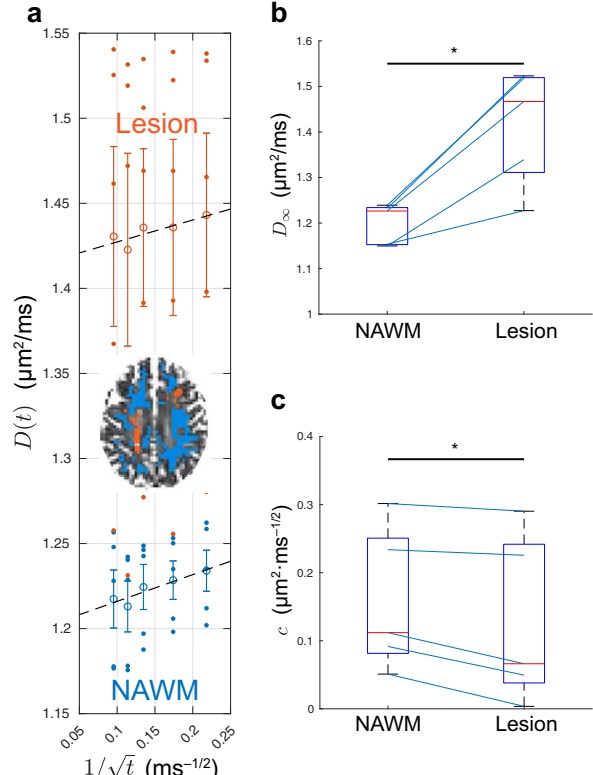

**Fig. 5 Time-dependent axial diffusivity $D(t)$ and fit parameters ($D_\infty$, $c$) estimated in brain lesions (red) and normal-appearing white matter (NAWM, blue) of five MS patients using monopolar PGSE. a** $D(t)$-values are plotted as points for each subject, along with corresponding mean values and error bars representing the SE over five subjects, and the corresponding fit to Eq. (1) confirms the $1/\sqrt{t}$ functional form of $D(t)$. **b** The bulk diffusivity $D_\infty$ along axons is significantly larger in MS lesions than that in NAWM. **c** The corresponding strength $c$ of restrictions along axons is smaller in MS lesions than in NAWM. In **b** and **c**, each patient is represented by a blue segment. The parameter differences between MS lesions and NAWM are compared by using one-sided Wilcoxon signed-rank test (*$P$-value < 0.05).

characterized by short-range disorder in 1d, with the dynamical exponent $\vartheta = 1/2$ for Eq. (1). This short-range disorder was confirmed by the power spectrum analysis of the actual shape of segmented myelinated axons in the 3d EM sample of mouse brain in this study, and it was also observed in a preliminary study[32] performing MC simulations within realistic axons segmented from a large 3d EM sample of human subcortical WM.

Furthermore, simulations in different microgeometries based on this EM sample allow us to disentangle the contributions of different microstructural features to the overall 1d structural disorder and reveal that the diffusivity and kurtosis time dependence along axons is dominated by caliber variations rather than axonal undulations. For example, in Supplementary Information, simulations of diffusion in fiber bundles composed of fibers without caliber variations, such as undulation-only fibers (geometry IV in Fig. 2a) or perfectly straight cylinders, demonstrate very small axial diffusivity time dependence along the main direction, even for highly dispersed case (Supplementary Fig. 3). Similarly, mitochondria have negligible impact on the time dependence, due to their low volume fraction (Supplementary Fig. 1). Yet, mitochondria are shown to correlate with axon caliber (Fig. 6) and hence could indirectly impact the time dependence, as discussed below for the MS pilot study.

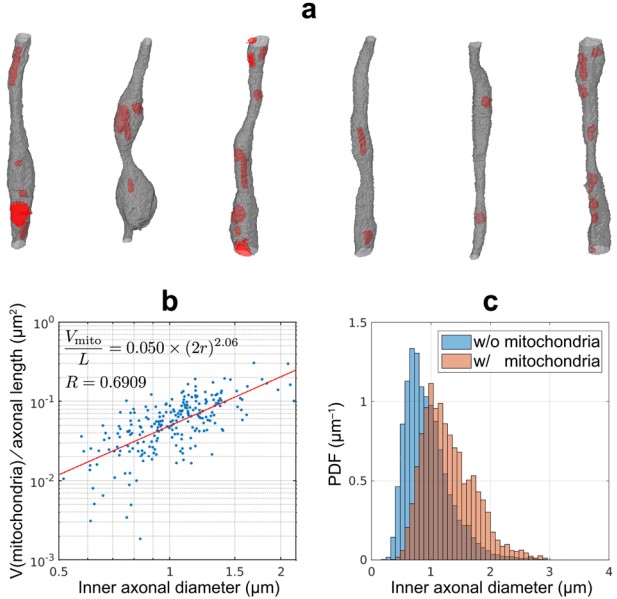

**Fig. 6 Mitochondria segmentation and characterization of the relation between mitochondria and axon caliber or diameter. a** Approximately 1300 mitochondria (red) in the 227 axons (gray) of Fig. 1a, b were manually segmented. **b** The inner axonal diameter $2r$ approximately correlates with each axon's mitochondrial volume per unit length ($V_{mito}/L$) via a quadratic function. **c** Axonal diameters in cross-sections where mitochondria are present (red) are significantly larger, compared to cross-sections where mitochondria are absent (blue) ($P$-value < 0.001).

Using PGSE dMRI in vivo in human brain, the power-law scaling (Eq. (1)) was found in all WM ROIs (Fig. 3a, b). In particular, in genu, the fit parameters of PGSE measurements (Table 3) are of the same order as those in the MC simulations for dispersion angles $\theta = 15°-30°$ (Table 2), which is consistent with the fiber dispersion $\simeq 20°$ observed in histology[23,33].

The fitted power-law parameters show similar patterns over different WM regions between subjects (Fig. 3a, b), and especially in CC composed of highly aligned axons, demonstrating the potential of clinical applications in the future, as discussed later. Admittedly, the regional variations in the bulk diffusivity $D_\infty$ and strength $c$ of restrictions are noisy for individual subjects; however, we are still able to observe the general trends across the CC for the average over all subjects: the trends relate remarkably well to the pattern of axonal density in CC observed in histology[34] and of axonal volume fraction in CC estimated via dMRI[17,35], as well as the higher spectrum of large axon diameters in the midbody according to ref. [34]. On the one hand, the high–low–high trend in $D_\infty$ in CC could be related to the pattern of axonal density in CC observed in histology, with the assumption that the axial diffusivity in IAS is larger than that in extra-axonal space[28]. On the other hand, the low–high–low trend in $c$ in CC could be related with the bead width and/or distance between local caliber maxima along individual axons. This observation cannot be supported or rejected by 2d histology and remains incompletely explained. For example, fiber bundles composed of (1) caliber-varying axons or (2) perfectly straight cylinders can have exactly the same 2d cross-sectional diameter distribution (Supplementary Fig. 3). Three-dimensional histology and analysis in different regions of CC are needed in the future to better understand our empirical observation of the trends across the CC.

In addition to in vivo PGSE measurements in human brain WM reported here, the power-law dependence has also been

reported using STE measurements in vivo in human brain WM[21] and ex vivo in the spinal cord WM[36], where the diffusion time is varied by changing the mixing time. Both studies reported somewhat stronger time dependence, as manifested by larger amplitude $c$ for the time dependence (cf. Table 2 of ref. [21] and Table 1 of ref. [36] as compared to current study Tables 2 and 3), a potential overestimation caused by water exchange between intra-/extra-axonal water (fast diffusion, long $T_1$, $T_2$ values) and myelin water (slow diffusion, short $T_1$, $T_2$ values) during the mixing time of the STE sequence[37]. Furthermore, in gray matter, the power-law dependence has been observed for the mean diffusivity using oscillating gradients in human brain[38] and rat brain[15,39], suggesting that the characteristics of short-range disordered restrictions to diffusion along axons and dendrites are a universal feature of neuronal tissue.

Conventionally, diffusion in WM has been modeled using the featureless stick model (reviewed by ref. [4]), thereby assuming Gaussian diffusion, corresponding to a negligible axial intra-axonal kurtosis. Here, however, based on realistic simulations, combined with theory and experimental verification, we conclude that the intra-axonal axial kurtosis is non-negligible at clinical diffusion times. Indeed, for $t = 20-100$ ms, the intra-axonal kurtosis along axons is ~0.7 for $\theta = 30°$ based on simulations (Fig. 2f), and ~0.8 for monopolar PGSE measurements in the human brain WM (Fig. 3c, d). The measured kurtosis in experiment is slightly larger than the intra-axonal kurtosis in simulations, likely due to the additional contribution of the extra-axonal space to the overall $K(t)$ in the measurement (Eq. (10b) in "Methods").

Simulations also demonstrated that the intra-axonal $K(t)$ increases with dispersion angle, especially for $\theta \gtrsim 30°$ (Fig. 2f), which can be understood by the corresponding increasing range of intra-axonal diffusivity values when projected to the fiber bundle's main direction, resulting in a larger contribution to the overall $K(t)$, i.e., the first right-hand-side term in Eq. (10b). Hence, the higher-order cumulants of the intra-axonal signal, including $K$, are very sensitive to the fiber dispersion (i.e., the functional form and the degree of orientation distribution) and should be incorporated in future biophysical models of dMRI in WM.

Besides the nominal (nonzero) value of the axial kurtosis, the observed time dependence of both $D(t)$ and $K(t)$ are nontrivial and should be considered in WM biophysical modeling. For $D(t)$ time dependence along axons, proportional to $\Delta(1/\sqrt{t}) \sim \Delta t \cdot t^{-3/2}$, it is negligible only when the time range $\Delta t$ is small (e.g., $\Delta t < 5$ ms), or the diffusion time is long (e.g., $t > 200$ ms). For $K(t)$, our simulations in IAS show 7% changes over the clinical time range $t = 20-100$ ms.

The observation of axon caliber variation and beading with non-invasive time-dependent dMRI calls for evaluating the role of this microstructural feature in pathology. In this work, we demonstrated altered diffusion time dependence along axons in WM lesions vs. NAWM of five MS patients (Fig. 5), with corresponding changes in the fit parameters that are potentially related to specific pathological changes. In particular, the increase in the bulk diffusivity $D_\infty$ along axons in MS lesions vs. NAWM (Fig. 5b) may suggest ongoing demyelination and axonal loss[40]. Although this observation alone has been reported before with dMRI[41–43], our results reveal, for the first time to our knowledge, that the diffusivity time dependence along WM axons, i.e., $c$ in Eq. (1), is smaller in MS lesions than that in NAWM, with $c \propto$ the correlation length $l_c$.[21] This observation is potentially indicative of an increase in mitochondria density, a feature of chronic demyelination documented from histology in axons and astrocytes of WM lesions[44]. As mitochondria and axon caliber are

shown to correlate (Fig. 6c)[45], an increase in mitochondria would shorten the correlation length $l_c$ that characterizes the distance between local maxima of the axon. Hence, the parameter $c$ potentially targets the specific pathology of mitochondria increase in MS. However, the exact relation of $c$ and restriction properties (e.g., their width and distance between them) is non-trivial and requires further explorations, for example, by solving the Fick-Jacobs equation for a caliber-varing fiber with randomly distributed beads along the fiber.

Conventional MRI methods (e.g., $T_2$-FLAIR) are well-known to distinguish MS lesions from NAWM and the distinction is typically attributed to demyelination[44]. Here, however, we aim to use MS lesion data to in vivo validate the strength $c$ of restrictions in Eq. (1) as a specific measure for changes in mitochondria: we demonstrate significant difference in $c$ between MS lesions and NAWM (Fig. 5c), and attribute it to an increase in mitochondria as a response to demyelination in MS lesions[44]. This observation may contribute to understanding the underlying pathological mechanisms taking place in MS lesion formation. In addition, our finding also suggests diffusion time dependence measurements as potential biomarker suitable for monitoring other pathologies presenting increased neurite beadings due to other mechanisms (rather than mitochondrial increase).

In addition to MS[46], axonal beading in WM has been observed in several other pathologies, such as traumatic brain injury (TBI)[47,48] and ischemic stroke[49]. Axonal varicosities, or axonal beading along axons, can be a pathological change caused by accumulation of transported materials in axonal swellings after TBI[47,48]; it has been observed that varicosities arise during dynamic stretch injury, caused by microtubule breakdown and partial transport interruption along axons. Furthermore, varicosities due to ischemic injury to WM axons can be caused by $Na^+$ loading of the axoplasm, which leads to a lethal $Ca^+$ overload through reversed $Na^+$-$Ca^+$ exchange[49]. Hence, the average distance between varicosities is potentially a biomarker for axonal injury in TBI and ischemia, facilitating evaluation of the effectiveness of treatment and rehabilitation services. As the average distance between varicosities along axons is of the order of 10 μm[47–49], much smaller than the resolution of most of the clinical imaging techniques, dMRI is the method of choice to estimate in vivo the pathological change of TBI[50,51] and of ischemic stroke[52]. In particular, time-dependent diffusion tensor imaging may enable the estimation of the correlation length of varicosities along axons, related to the average distance between varicosities, a potential biomarker for monitoring TBI and ischemic stroke patients.

Besides beading in WM, the ubiquitous $1/\sqrt{t}$ time dependence along neurites in gray matter[15,39,53] suggests possible applications in other neurodegenerative diseases. For instance, reduced density of axonal varicosities was observed in the human superior frontal cortex of mild to moderate Alzheimer disease[54]; decreased dendritic spine density was observed in the human prefrontal cortex of Schizophrenia[55]; and an increased density of axonal varicosities was observed in injured dopaminergic neurons in the rat substantia nigra, an animal model of Parkinson's disease[56]. The ability to evaluate restriction changes along neurites opens a door to monitoring the progression and therapy response of these diseases.

Thanks to recent advances in 3d EM[57], our work, for the first time to our knowledge, demonstrate the feasibility to employ EM-derived microstructure as numerical phantoms for realistic 3d simulations. By fully controlling the microgeometry of numerical phantoms, MC simulations provide complete flexibility to evaluate the influence of different microstructural features. Here, they were employed to elucidate that the time-dependent diffusion signal along axons mainly originates from the caliber variations,

with the contributions from mitochondria and axonal undulations having relatively small effects.

While we demonstrate here the value of realistic simulations as a validation tool, one can think of extending this approach further to study the sensitivity of MRI to microstructure. Larger EM samples[58] would be needed to enable diffusion simulations at longer diffusion times. For the EM sample used in the current study, the maximal axon length $L \sim 18$ μm corresponds to a length-related correlation time $\tau_L = L^2/(2D_0) \simeq 80$ ms for $D_0 \sim 2$ μm²/ms, which sets the maximum feasible diffusion time for the simulation. This maximum time $\sim\tau_L$ also prevents us from validating the relation of power-law scaling in $D(t)$ and $K(t)$, i.e., $\Delta D(t)/D_\infty = \Delta K(t)/2$[68], since the power-law scaling in $K(t)$ happens at longer diffusion times. Furthermore, we only focused on the intra-axonal geometry of myelinated axons in WM. Although the contribution of extra-axonal space is non-negligible, extra-axonal signals are relatively smaller than the intra-axonal ones because of the shorter $T_2$ in extra-axonal space and long echo time applied in experiments[28]. For the diffusivity time dependence along the fiber bundle, we expect that the diffusivity time dependence in extra-axonal space is similar to that in IAS, as water molecules experience similar beading arrangement in either intra- or extra-axonal spaces. Faithfully segmenting and simulating the diffusion in the extra-axonal space is needed to understand how robust the observed power-law is with increasing dispersion. In addition, other structures, such as unmyelinated axons, glia cell, and blood vessels, may have nontrivial contributions to the (time-dependent) diffusion signal and can be added to the numerical microgeometry. Ultimately, a large human EM sample[32] (comparable to MRI voxel size), prepared with extra-cellular space preserving technique if possible, would provide the most representative numerical phantom to the human tissue microstructure after fully segmenting all the cells inside the sample.

Finally, although the proposed framework here focuses on performing MC to model diffusion in realistic WM microstructure, it can also be applied to gray matter, or tissue samples with pathology. In addition, the framework can be extended to include other MR contrast mechanisms, e.g., magnetization transfer, mesoscopic susceptibility[59], $T_1$ and $T_2$ relaxation[60], and water exchange[61], thereby facilitating the exciting ability to validate non-invasive MR-based tissue microstructural mapping.

## Methods

All procedures performed in studies involving animals were in accordance with the ethical standards of New York University School of Medicine. All mice were treated in strict accordance with guidelines outlined in the National Institutes of Health Guide for the Care and Use of Laboratory Animals, and the experimental procedures were performed in accordance with the Institutional Animal Care and Use Committee at the New York University School of Medicine. All procedures performed in studies involving human participants were in accordance with the ethical standards of New York University School of Medicine. All protocols were approved by the local institutional review board (New York University School of Medicine). Informed consent was obtained from all individual participants included in the study.

**EM and IAS segmentation**. The brain tissue from a female 8-week-old C57BL/6 mouse's genu of CC was processed and analyzed with a scanning EM (SEM) (Zeiss Gemini 300 SEM with 3View). Part of the data was discarded due to unstable quality, leading to a volume (Fig. 1a) of $36 \times 48 \times 20$ μm³. To segment long axons passing through all slices, we employed a simplified seeded region growing algorithm[23,62,63]. The segmented axons (Fig. 1b) shorter than 20 μm were discarded, leading to 227 long axons (≥20 μm in length). More details were reported in our previous work[23].

The IAS segmentation was down-sampled into a voxel size of $(0.1$ μm$)^3$. The effect of orientation dispersion was controlled by subsequently realigning axons along the $z$-axis (Fig. 2a). The aligned axons were truncated at both ends by 1 μm to avoid oblique end faces, resulting in axons of about 18 μm in length.

**Mitochondria density affects inner axonal diameter**. To evaluate the influence of mitochondria on the axon caliber variation and on the diffusion time

dependence, we manually segmented ~1300 mitochondria in 227 axons (Fig. 6a). For individual axons, their inner diameter is found to correlate with the mitochondrial volume per unit length via a quadratic function (Fig. 6b), similar to the observation in ref. [64]. In addition, the axonal diameters calculated based on cross-sections with and without the presence of mitochondria are $1.29 \pm 0.43$ µm ($n = 13,653$) and $0.94 \pm 0.38$ µm ($n = 31,747$), respectively (Fig. 6c), indicating that the presence of mitochondria in IAS corresponds to larger axonal diameters ($P$-value < 0.001) and in agreement with a previous histological study in human and non-human primate retinas[45].

**MC simulations in realistic microstructure**. MC simulations of random walkers were implemented in CUDA C++ for diffusion in a continuous space. Walkers ($2.27 \times 10^9$ in total) were employed inside 3d segmentations of 227 IASs, with $1 \times 10^7$ walkers per IAS. The walker encountering the cell membrane is elastically reflected or permeates through the membrane based on a permeation probability for highly permeable membranes[65], $P_{1 \to 2} = \min(1, \sqrt{D_2/D_1})$ with intrinsic diffusivities $D_1$ and $D_2$ in compartments 1 and 2. The top and bottom faces of each IAS binary mask, artificially made due to the length truncation, were extended with its relative copies (mirroring boundary condition) to avoid geometrical discontinuity in diffusion simulations[24].

Each particle diffused over $5 \times 10^5$ steps with a duration $\delta t = 2 \times 10^{-4}$ ms and a length $\sqrt{6D_a \delta t} = 0.049$ µm for each step in IAS and $\sqrt{6D_m \delta t} = 0.013$ µm in the mitochondria, where the intrinsic diffusivity, $D_a = 2$ µm²/ms in IAS and $D_m = 0.13$ µm²/ms in the mitochondria, is taken to agree with recent in vivo experiments[10,11] and previous in vitro study[29]. Maximal diffusion time in simulations is $t = 100$ ms. Total calculation time was ~4 days on ~20 NVIDIA Tesla V100 GPU on the NYU Langone Health BigPurple high-performance computing cluster.

The $i$-th axon's moment tensors $\langle x_{j_1} x_{j_2} \rangle_i$ and $\langle x_{j_1} x_{j_2} x_{j_3} x_{j_4} \rangle_i$ are calculated in the axon's frame of reference based on the simulated diffusion displacement vector $\mathbf{x}$ (with the component $x_{j_1}$, $j_1 = 1, 2,$ or 3)[66,67], and their projections yield the axon's apparent diffusivity $D_i(t, \hat{\mathbf{n}})$ and apparent kurtosis $K_i(t, \hat{\mathbf{n}})$ in the direction $\hat{\mathbf{n}}$ (with the component $n_{j_1}$)[66,67]:

$$D_i(t, \hat{\mathbf{n}}) = \frac{\langle s^2 \rangle_i}{2t}, \tag{8a}$$

$$K_i(t, \hat{\mathbf{n}}) = \frac{\langle s^4 \rangle_i}{\langle s^2 \rangle_i^2} - 3, \tag{8b}$$

where

$$\langle s^2 \rangle_i = n_{j_1} n_{j_2} \langle x_{j_1} x_{j_2} \rangle_i,$$
$$\langle s^4 \rangle_i = n_{j_1} n_{j_2} n_{j_3} n_{j_4} \langle x_{j_1} x_{j_2} x_{j_3} x_{j_4} \rangle_i,$$

and the summation over the pairs of repeating indices is implied.

To simulate the effect of any fiber orientation dispersion, we draw an axon's direction $\hat{\mathbf{n}}'$ from the orientation distribution and then calculate the axon's apparent diffusivity and apparent kurtosis along the $z$-axis (fiber bundle) direction by using $D_i(t, \hat{\mathbf{n}})$ and $K_i(t, \hat{\mathbf{n}})$ in the direction $\hat{\mathbf{n}} = 2(\hat{\mathbf{n}}' \cdot \hat{\mathbf{z}})\hat{\mathbf{z}} - \hat{\mathbf{n}}'$. This is similar to the reflection of light, with the incident light along $-\hat{\mathbf{n}}'$ falls on the surface normal to $\hat{\mathbf{z}}$, and is reflected along $\hat{\mathbf{n}}$. It is then straightforward to calculate the overall $D(t)$ and $K(t)$ using Eq. (10) below.

**Ensemble averaging over axons**. The dMRI signal from many axons can be approximated by the cumulant expansion[3,66]

$$S(b, t) \simeq e^{-bD(t) + \frac{1}{6}b^2 D^2(t)K(t) + \mathcal{O}(b^3)}$$
$$= \sum_i f_i \cdot e^{-bD_i(t) + \frac{1}{6}b^2 D_i^2(t)K_i(t) + \mathcal{O}(b^3)}, \tag{9}$$

where $D(t)$ and $K(t)$ are overall diffusivity and kurtosis, and $D_i(t)$ and $K_i(t)$ are diffusivity and kurtosis of individual axons with volume fractions $f_i$, such that $\Sigma_i f_i \equiv 1$. Expanding Eq. (9) up to $b^2$, we obtain[66,68]

$$D(t) = \langle D_i(t) \rangle \equiv \sum_i f_i \cdot D_i(t), \tag{10a}$$

$$K(t) = \frac{1}{D^2(t)} \sum_i \left[ 3f_i \cdot (D_i(t) - D(t))^2 + f_i \cdot D_i^2(t)K_i(t) \right]. \tag{10b}$$

Equation (10a) yields that the overall $D_\infty$ and $c$ entering Eq. (1) are given by the volume-weighted averages of the corresponding parameters of the individual axons,

$$D_\infty \equiv \langle D_{i,\infty} \rangle, \quad c \equiv \langle c_i \rangle. \tag{11}$$

Throughout, we use the time interval $t = 20$–$80$ ms to fit $D_{i,\infty}$ and $c_i$ from MC simulations of individual axons, i.e., fitting Eq. (1) to $D_i(t)$, and employ these parameters to predict the axial diffusivity $D(t)$ of all axons in Eq. (1) and Eq. (11). The maximal diffusion time used for fitting is bounded by the axonal length of the EM substrate $L \sim 18$ µm: $L^2/(2D_0) \simeq 80$ ms for $D_0 = 2$ µm²/ms.

Considering a fiber bundle with the orientation dispersion, the diffusion displacement within an axon (dispersed along $\theta_i$) is generally along the axon due to

its thin size. Its projection to the fiber bundle's main direction leads to a contribution to the second order cumulant $\langle s^2 \rangle_i \propto \cos^2 \theta_i$ along the fiber bundle. As a result, the overall diffusivity and corresponding parameters are given by

$$\frac{D(t)}{D(t)|_{\theta=0}} = \frac{D_\infty}{D_\infty|_{\theta=0}} = \frac{c}{c|_{\theta=0}} \simeq \langle \cos^2 \theta \rangle. \tag{12}$$

However, for a highly dispersed fiber bundle (e.g., $\theta = 45°$), some axons are oriented roughly perpendicular to the fiber bundle's main direction; these axons' radial diffusivity $\propto 1/t$ can be projected to the main direction, resulting in a small contribution to the overall axial diffusivity $D(t)$[21], biasing the estimate of $c$. To account for this contribution, a correction term is added to the overall $D(t)$ in Eq. (1):

$$D(t) \simeq D_\infty + c \cdot \frac{1}{\sqrt{t}} + c' \cdot \frac{1}{t}, \tag{13}$$

where $c'$ is related with caliber variation[20] and undulation[69].

**Relation of relative caliber variation and relative diffusivity variation**. In Fig. 2d, the metric specifying the axonal shape, the coefficient of variation of radius $\mathrm{CV}(r) = \sigma_r / \langle r \rangle$ ($\sigma_r$ is the SD and $\langle r \rangle$ is the mean radius), and the relative diffusivity variation $\zeta \equiv (D_0 - D_{i,\infty})/D_{i,\infty}$[70] highly correlate with each other. Note that, $\mathrm{CV}(r)$ is calculated solely based on axons' 3d microgeometry; in contrast, the relative diffusivity variation is estimated based on simulation results. To explain this observation, we derive a relation to link the two metrics.

Our argument is based on the coarse-graining of 1d axonal microstructure by diffusion[4,15]. When the diffusion length $L_d(t)$ grows beyond the correlation length of caliber variations, all the effective 1d diffusion physics is represented by a 1d coarse-grained diffusion coefficient $D(z)$ varying in space on the scale $L_d(t)$. For sufficiently large $L_d(t)$ (long $t$), the local fluctuations $\delta D(z) = D(z) - \overline{D}$ become small, i.e., $|\delta D(z)| \ll \overline{D}$, where $\overline{D}$ is the average of $D(z)$ along the axon. In particular, the local fluctuation of the coarse-grained local 1d diffusivity $\delta D(z) \simeq (\partial \overline{D}/\partial \bar{n})\delta n(z)$ is proportional to the local fluctuation of restriction density $\delta n$, with $\bar{n}$ the mean density[15]. It is then straightforward to calculate each individual axon's bulk diffusivity $D_{i,\infty}$, given by[70]

$$\frac{1}{D_{i,\infty}} = \left\langle \frac{1}{D_i(z)} \right\rangle_z \simeq \frac{1}{\overline{D}} \left[ 1 + \frac{\langle (\delta D)^2 \rangle_z}{\overline{D}^2} \right]$$
$$\simeq \frac{1}{\overline{D}} \left[ 1 + \left( \frac{\partial \ln \overline{D}}{\partial \ln \bar{n}} \right)^2 \frac{\langle (\delta n)^2 \rangle_z}{\bar{n}^2} \right],$$

simplified as

$$\frac{\overline{D} - D_{i,\infty}}{D_{i,\infty}} \propto \frac{\langle (\delta n)^2 \rangle_z}{\bar{n}^2} \tag{14}$$

to the lowest order in $\delta n$. Above, we neglected the third and higher orders of $\delta n$ and so this derivation is by construction perturbative and valid for small $\zeta$ and CV($r$).

The CSA variation $A(z)$ along an axon can be expressed as the convolution of restriction density $n(z)$ and 3-dimensional shape function of a restriction $v(z)$, i.e., $A(z) = n(z) * v(z)$, or in the Fourier domain, $A(k_z) = n(k_z)v(k_z)$. The coarse-grained density fluctuation $\delta n(k_z)$ at scales much longer than the mean restriction width $\bar{l}$, corresponding to $k_z \cdot \bar{l} \ll 1$, causes the corresponding fluctuation

$$\delta A(k_z) = \delta n(k_z)v(k_z) \simeq \delta n(k_z)v_0, \quad v_0 = v|_{k_z=0} \sim \varphi \cdot \frac{\overline{A}}{\bar{n}}.$$

Here, $v_0$ is the restriction strength (e.g., single bead volume), and $\varphi$ is the volume fraction of restrictions. Hence, when $\varphi \sim$ const, $\delta A(k_z)/\overline{A} \sim \delta n(k_z)/\bar{n}$, or

$$\frac{\delta n(z)}{\bar{n}} \sim \frac{\delta A(z)}{\overline{A}} \propto \frac{\delta r}{\langle r \rangle},$$

as $\delta r = r - \langle r \rangle$, $\delta A \sim \langle r \rangle \cdot \delta r$, and $\overline{A} \sim \langle r \rangle^2$. Substituting into Eq. (14) and approximating the local average diffusivity by the free diffusivity $\overline{D} \simeq D_0$ (no restrictions for $\delta n = 0$), we obtain

$$\zeta \equiv \frac{D_0 - D_{i,\infty}}{D_{i,\infty}} \propto \frac{\langle (\delta A)^2 \rangle_z}{\overline{A}^2} \propto \frac{\langle (\delta r)^2 \rangle}{\langle r \rangle^2} = \mathrm{CV}^2(r), \tag{15}$$

which is demonstrated by plotting $\zeta$ versus $\mathrm{CV}^2(r)$ in Fig. 2d, where the correlation coefficient = 0.8917 is high and a small intercept = 0.09 verifies this simple relation. We note that Eq. (15) has been derived for small $\zeta$ and CV($r$), and the scatter close to the origin in Fig. 2d is indeed much closer to the straight line. Similarly, a decrease of axial diffusivity due to increased amplitude of periodically arranged beads was numerically observed in ref. [71].

**In vivo MRI of healthy subjects**. dMRI measurements were performed on five healthy subjects (four males/one female, 21–32 years old) using a monopolar PGSE sequence provided by the vendor (Siemens WIP 919B) on a 3T Siemens Prisma scanner (Erlangen Germany) with a 64-channel head coil. For each subject, we varied the diffusion time $t = [22, 28, 34, 40, 50, 60, 70, 80, 90, 100]$ ms and fixed the diffusion gradient pulse width $\delta$ at 15 ms. For each scan, we obtained three $b = 0$

non-diffusion-weighted images (DWIs) and 62 DWIs of $b$-values $b = [0.4, 1, 1.5]$ ms/μm$^2$ along [12, 20, 30] gradient directions for each b-shell, with an isotropic resolution of $(3 \text{ mm})^3$ and a field-of-view (FOV) of $210 \times 204$ mm$^2$. The whole brain volume was scanned within 30 slices, aligned parallel to the anterior commissure (AC)–posterior commissure (PC) line. GRAPPA with acceleration factor = 2 and multiband with acceleration factor = 2 were used. All scans were performed with the same TR/TE (repetition time/echo time) = 4000/139 ms. Total acquisition time is ~60 min for each subject. In the main text, we focus on this dataset. The data of 10 additional subjects scanned with a smaller voxel size, exhibiting similar outcomes, are shown in Supplementary Information.

Our image processing DESIGNER pipeline is based on ref. [72] and includes five steps: denoising, Gibbs ringing elimination, Eddy current and motion correction, and Rician noise correction. For each voxel, we fitted dMRI data to the diffusion and kurtosis tensor using weighted linear least square[73], and calculated eigenvalues of the diffusion tensor (in the order of $\lambda_1 \geq \lambda_2 \geq \lambda_3$) and the fractional anisotropy (FA) accordingly[74]. Experimental axial diffusivity is defined by $D \equiv \lambda_1$ and experimental axial kurtosis is defined as the apparent kurtosis along the principal axis of the diffusion tensor.

Each subject's mean FA map, averaged over all diffusion time points, was registered to FSL's (FMRIB Software Library) standard FA map with FMRIB's (Functional MRI of the Brain) linear and non-linear registration tools (FLIRT, FNIRT)[75,76]. We retrieved the transformation matrix (FLIRT) and the warp (FNIRT) to inversely transform Johns Hopkins University (JHU) DTI-based WM atlas ROIs[77] to the individual space. Cerebrospinal fluid (CSF) mask was segmented by FSL, FAST[78] and expanded by 1 voxel to exclude WM voxels close to CSF. We focused on main WM tracts, such as anterior corona radiata, posterior corona radiata, superior corona radiata, anterior and posterior limb of the internal capsule, genu, midbody, and splenium of the CC.

To further discuss the variation of tissue properties in CC, we divided CC ROIs defined in JHU DTI atlas into nine sub-regions in total (Fig. 4a), such as G1, G2, G3 for the genu, B1, B2, B3 for the midbody, and S1, S2, S3 for the splenium. The nine sub-regions are then co-registered and transformed to individual subject's space by using FSL.

**In vivo MRI of multiple sclerosis patients**. The dMRI measurements were performed on five MS patients (five females, 32–48 years old) using a monopolar PGSE sequence provided by the vendor (Siemens WIP 511E) on a 3T Siemens Prisma scanner (Erlangen Germany) with a 64-channel head coil. For each subject, we varied the diffusion time $t = 21$–110 ms and fixed the diffusion gradient pulse width δ at 15 ms. For each time point, we obtained three $b = 0$ non-DWIs and DWIs of $b = 0.5$ ms/μm$^2$ along 30 gradient directions, with an isotropic resolution of $(3 \text{ mm})^3$ and an FOV of $222 \times 222$ mm$^2$. A slab of the brain volume was scanned within 15 slices, aligned parallel to the AC–PC line. All scans were performed with the same TR/TE = 4200/150 ms. Total acquisition time of DWIs is ~15 min for each subject.

Sagittal 3d MPRAGE (magnetization-prepared rapid gradient echo) brain images were acquired with an isotropic resolution of $(1 \text{ mm})^3$, an FOV of $256 \times 256$ mm$^2$, TR/TE = 2100/2.72 ms, and inversion time = 900 ms. Axial FLAIR brain images were acquired with an anisotropic resolution of $0.6875 \times 0.6875 \times 5$ mm$^3$, an FOV of $220 \times 220$ mm$^2$, TR/TE = 9000/90 ms, and inversion time = 2500 ms.

The image processing pipeline was the same as that in healthy subjects. MS patients' WM lesions were manually segmented by identifying hyper-intensity regions in FLAIR images. The segmented lesions were further transformed to the DWI space by using FLIRT and FNIRT[75,76]. The NAWM was segmented in MPRAGE images by using FAST[78] and transformed into the DWI space. To avoid partial volume effect, we excluded voxels close to MS lesions and CSF by expanding the mask of lesions and CSF by one voxel. An example of ROIs of MS lesions and NAWM is shown in Fig. 5a.

**Statistics and reproducibility**. The normality of distributions of inner axonal diameters in cross-sections with or without the presence of mitochondria was tested by using Anderson–Darling test, with a null hypothesis of normal distribution at 0.05 significance level; the null hypothesis was rejected for both diameter distributions with $P$-values < 0.001. Further, the two diameter distributions were compared using one-sided Wilcoxon sum-rank test, with the null hypothesis that axonal diameters with the presence of mitochondria are not larger than those without. The significance level is 0.05.

Eigenvalues and axial diffusivity were calculated voxel by voxel and averaged over each ROI. To evaluate the strength of axial diffusivity time dependence in healthy subjects, we assumed that $D(t)$ is a linear function of $1/\sqrt{t}$ based on Eq. (1) and calculated $P$-values with the null hypothesis of no positive correlation (one-sided test). Both the time-dependent parameters ($D_\infty$, $c$) in WM lesions and NAWM of MS patients did not pass the normality test (Anderson–Darling test) and were therefore compared using a paired one-sided Wilcoxon signed-rank test. For bulk diffusivity $D_\infty$, the null hypothesis is that $D_\infty$ in lesions is not larger than that in NAWM; for strength $c$ of restrictions, the null hypothesis is that $c$ in lesions is not smaller than that in NAWM. The significance level is 0.05.

In this study, we chose one-tailed non-parametric test as we specifically hypothesized a decrease in the strength $c$ of restrictions in MS lesions as compared with NAWM, and an increase in the bulk diffusivity $D_\infty$. Indeed, as (1) the inner

axonal diameters in cross-sections with the presence of mitochondria are larger than those without based on the previous histological study[45] and Fig. 6c, and (2) mitochondria density is increased in MS lesions due to demyelination[44], more variation in axon caliber is expected in lesions with a corresponding decrease in $c$. Similarly, we expect that $D_\infty$ in MS lesions is larger than that in NAWM due to demyelination[40]. We would like to note that the MS data is rather exploratory due to the small sample size ($n = 5$), which increases the risk of type 2 errors. In addition, the smallest possible $P$-value of a one-sided Wilcoxon signed-rank test with $n = 5$ is 0.03125, which provides a lower-bound for the $P$-value in this study.

**Reporting summary**. Further information on research design is available in the Nature Research Reporting Summary linked to this article.

## Data availability
The SEM data and axon segmentation can be downloaded on our web page (http://cai2r. net/resources/software). All human brain MRI data for this study are available upon request. Data underlying Figs. 1–6 are provided as Supplementary Data.

## Code availability
The source codes of Monte Carlo simulations can be downloaded on our github page (https://github.com/NYU-DiffusionMRI).

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

## Acknowledgements

We thank Thorsten Feiweier for developing advanced diffusion WIP sequence and Bigpurple High Performance Computing Center of New York University Langone Health for numerical computations on the cluster. Research was supported by the National Institute of Neurological Disorders and Stroke of the NIH under award number R01 NS088040 and R21 NS081230, and by the National Institute of Biomedical Imaging and Bioengineering (NIBIB) of the NIH under award number U01EB026996, and was performed at the Center of Advanced Imaging Innovation and Research (CAI2R, www.cai2r. net), an NIBIB Biomedical Technology Resource Center (NIH P41 EB017183). Figure 1a, b is adapted by permission from Springer Nature: Brain Structure and Function. Along-axon diameter variation and axonal orientation dispersion revealed with 3D electron microscopy: implications for quantifying brain white matter microstructure with histology and diffusion MRI. Lee et al. Copyright, 2019[23].

## Author contributions

H.H.L., D.S.N., and E.F. designed research. H.H.L., D.S.N., and E.F. designed simulations. H.H.L. performed simulations. H.H.L. and D.S.N. developed theory. H.H.L., D.S.N. and E.F. designed experiments. H.H.L., A.P. and E.F. performed experiments. H.H.L. and S.L. K. performed segmentations. H.H.L. analyzed data. D.S.N. and E.F. supervised the project. H.H.L., D.S.N. and E.F. wrote the paper.

## Competing interests

The authors declare no competing interests.
