## [Peer Review File · Communications Biology]

Reviewers' comments:

Reviewer #1 (Remarks to the Author):

The research reported here combines meticulous theoretical analysis, rigorous MC simulations, and clinically relevant diffusion MRI experiments to propose and validate a power-law relation for the time-dependence of along-axon diffusion of water molecules in the intra-axonal space. The power-law relation suggests a role of short-range structural disorder in restricting along-axon diffusion, which the authors convincingly connect to axonal beading vis-a-vis axonal undulations. A particularly striking insight is the correlation between axonal beading and mitochondrial density, which leads the authors to suggest that the scaling factor (restriction amplitude, c) could serve as a biomarker for detecting MS lesions, which are characterized by an increased mitochondrial density along axons. Overall, the reported work is novel and rigorous in execution. It is likely to generate a significant impact in the field in at least three ways: (1) shifting the paradigm away from the infinitely thin stick model of Gaussian intra-axonal diffusion (2) generating a noninvasive biomarker for detecting neurodegenerative diseases and traumatic brain injuries (3) stimulating follow-up efforts to connect diffusion MRI readout with cellular level pathology, thereby bridging at least 3 orders of magnitude in space. I have no reservations in recommending the manuscript for publication, following a few minor revisions as suggested below:

1. Page 3, column 2, 2nd paragraph - "focal swellings (caliber maxima)" is repeated twice.
2. The authors comment on the possible use of the scaling factor, c as a biomarker for MS lesions. As is well known (and evident in Figure 6(b)), D_{∞} is already a viable biomarker for distinguishing MS lesions from healthy/normal-appearing white matter. Could the authors amplify their discussion on how a second biomarker for MS might be useful? For example, in what ways could using c as a biomarker benefit diagnosis differently from using D_{∞} ? Also, is the difference in c values between NAWM and MS lesions sufficient to enable reliable discrimination or would the risk of type II errors be significant?
3. I am curious as to why the authors chose to perform 1-tailed non-parametric tests rather than opt for the more conservative 2-tailed version?

Reviewer #2 (Remarks to the Author):

MAJOR

This paper explores the possibility of detecting local changes in axon diameter with dMRI and it is therefore quite along the lines of the most advanced contemporary studies aiming to improve the resolution of dMRI measurements of white matter parameters.

The interesting approach is to explore diffusion signals to gradients applied longitudinally rather than transversally (as it is usually done) and to assess diffusion over time. The authors are aware of the local changes in axon diameter that they have demonstrated in previous study of the mouse CC. From the same work they also have extracted estimates of axonal misalignment (here called dispersion).

They were therefore in a good position to test the consequences of 1. Local diameter variation due to swelling or mitochondria and 2. Axonal misalignment. By the construction of sophisticated mathematical modeling and simulations they come to the important conclusion that the local variation in axial diffusivity are due to the local axon diameter changes rather than axonal bending or misalignment.

I am not fully convinced by this conclusion, in spite of the great interest and beauty of the work.

One of the problems is that the mouse CC provided estimates of local misalignment over a 20 μ m slice while the dMRI signal is extracted from a far thicker section which will certainly produce and estimation over-fitting on the Monte-Carlo simulations towards local inhomogeneities of the few extracted samples.

The second problem is that the mouse CC does not provide appropriate guidance for the work in humans since axon diameters are much more uniform in rodents than in primates.

In conclusion, why is the radial diffusivity function much steeper for the midbody of CC than for other CC districts? Can this be due to some special structural feature of this sector of CC, in particular. the greater variation in axon diameters, with very thick and thin axons interspersed?

Indeed, the authors should demonstrate that the variation in longitudinal diffusivity are not due to changes in diameter across axons rather than along axons.

Finally, a concern about the conclusions reached on synthetic data is that they cannot be directly generalized to human dMRI data, since the extra-axonal space may be a concomitant constituent of the time dependency measured in real data.

MINOR

What is c in eq 1 ? say it right away.

Reviewer #3 (Remarks to the Author):

This is a manuscript by an eminent group that makes a potentially substantial contribution to how diffusion MRI data are interpreted within white matter. The authors demonstrate a diffusion time dependence of quantitative parameters derived from a technically challenging set of diffusion MRI measurements. In order to interpret their experimental data, the authors perform simulations of water diffusion within environments resembling a white matter fiber. 3D EM reconstructions of axons are used to provide input to the simulations, and a series of elegant analyses are performed, which the authors ultimately use to assert that variation in axon diameter along its axis, consistent with "axon beading", is responsible for the diffusion time dependence. Other potential factors, such as undulations in the axon axis, and the presence of mitochondria, are reasoned to be less likely by the authors. These results are placed within a context of an extensive theoretical foundation the authors have developed over recent years.

The majority of my comments are related to how the manuscript is presented. Only the first comment below potentially bears on the soundness of the conclusions.

1. Mitochondria were explicitly segmented in the 3D EM data. In order to incorporate the potential effects of mitochondria in simulations of water diffusion, they were modeled as structures with "fully permeable" membranes, but distinct from the cytoplasm in that the water T2 is 20 ms (rather than the T2 of 80 ms assumed for water in the cytoplasm). Work by Cerdan et al. is cited to support these choices of physical parameters. However, another characteristic of mitochondria described by the Cerdan group is its 15-30 times larger apparent viscosity than that of axoplasm, which leads to "reductions of more than one order of magnitude of the diffusion coefficient of water." The authors do not seem to incorporate this difference between cytoplasmic and mitochondrial water in their simulations. This could be an important confound, given the correlation between mitochondrial location and regions of locally increased axon diameter.

2. It would be helpful to see a figure that combines the experimental data with the data from simulations. Given the different behavior for different WM structures (Figure 4) and the different curves as parameters are varied in the simulations (Figures 2 and 3), a combined figure could help the reader appreciate how well the theory and experiment match, or how the WM structures differ

from each other.

3. The "pilot study" of Figure 6 does not contribute to this manuscript's primary focus, to establish the role of variation in axon diameter in diffusion MRI measurements. I am in favor of removing this section, as it raises many distracting questions related to segmentation, the role of variation between WM structures, contributions of various other characteristics of WM pathology, etc.

4. In the results section devoted to fiber dispersion, a range of Watson distribution concentration parameters was explored. The concentration parameter obtained from the axons in the 3D EM data should be provided here for context.

5. The last 4 paragraphs of the Introduction are unnecessary. A succinct summary might be helpful in the last paragraph of the Introduction, but detailed summaries such as the one provided is more common in engineering journals than they are (I think) in Communications Biology papers. Similarly, the 3rd paragraph of the Discussion is not needed. The Discussion section subheadings serve the purpose.

6. It might be helpful to provide 1-2 sentence transitions at the beginnings of paragraphs that review the different physical parameters (pages 5,6). As examples, the reason for evaluating the effect of fiber dispersion could be provided at the beginning of the paragraph that describes Figure 3. This and the next paragraph could provide a 1-sentence reminder of the information related to the parameter "c". The next paragraph (beginning on page 6) abruptly begins a discussion of $K(t)$, which, the reader might want to be reminded, refers to the time dependence of kurtosis.

7. On page 5, the text following the roman numeral II states that T2 contrast between axoplasm and mitochondria is removed by making $T2a = T2m$, but the numerical value is not provided. Presumably it is 80 ms, but the number should be given.

8. On pages 10-11, it was difficult to find where the meaning of the parameters $n_{i,j}$ and $x_{i,j}$ are defined.

28 March 2020

Probing axon caliber variations and beading with time-dependent diffusion MRI

We sincerely thank all reviewers for their time, effort and suggestions for improving the manuscript. We have attempted to take the reviewers' suggestions into account and addressed all raised comments.

Most importantly, we have performed additional Monte Carlo simulations. Using a micro-geometry composed of densely-packed highly-aligned straight cylinders, each with no caliber variation but with wide diameter distribution across cylinders, we show that the diffusivity time-dependence along fibers is not caused by the diameter variation across fibers. Furthermore, we consider the low diffusivity in mitochondria (due to its low viscosity) in MC simulations, which are completely re-implemented to account for this change. Finally, we address the raised concern of our small mouse brain EM sample potentially being *under-representative* (phrased as an *overfitting* problem) by showing that the confidential preliminary results of our ongoing project (human brain EM sample with a volume of 2 x 3 x 0.2 mm in the attached file "ISMRM_2020_abstract_278.pdf") are consistent with the results of this study.

As a reference, we have enclosed the manuscript in which the text additions supplied in response to the Reviewer critiques and suggestions are shown in **red**. Below, we copy each of the comments of the reviewer in *italic* font followed by our response in regular font.

Reviewer #1

I have no reservations in recommending the manuscript for publication, following a few minor revisions as suggested below:

MINOR

R1.1. Page 3, column 2, 2nd paragraph - "focal swellings (caliber maxima)" is repeated twice.

Thank you for noticing, the repeated words are deleted now.

R1.2. The authors comment on the possible use of the scaling factor, c as a biomarker for MS lesions. As is well known (and evident in Figure 6(b)), D_{inf} is already a viable biomarker for distinguishing MS lesions from healthy/normal-appearing white matter. Could the authors amplify their discussion on how a second biomarker for MS might be useful? For example, in what ways could using c as a biomarker benefit diagnosis differently from using D_{inf} ? Also, is the difference in c values between NAWM and MS lesions sufficient to enable reliable discrimination or would the risk of type II errors be significant?

We agree with the reviewer that the bulk diffusivity D_{∞} already shows significant differences between MS lesions and NAWM, and has been proposed as a biomarker for *demyelination* and neuronal degeneration. In addition, other MRI modalities (e.g. T₂-weighted) are able to distinguish MS lesions from NAWM. Rather than aiming to propose another biomarker for MS lesions, we used MS data in this study to demonstrate that the strength of restrictions c (obtained from time-

dependence diffusion MRI measurement) is significantly different between lesions and NAWM, and attribute this difference to the well-known *increase in mitochondria*, as a response to *demyelination* in MS lesions (Witte, et al., *Trend in Molecular Medicine*, 2013, doi:10.1016/j.molmed.2013.11.007). This observation may contribute to understanding the underlying pathological mechanisms taking place in MS lesion formation. In addition, our finding also suggests diffusion time-dependence measurements as potential biomarker suitable for monitoring other pathologies, such as traumatic brain injury and ischemic stroke, presenting increased neurite beadings due to other mechanisms (rather than mitochondrial increase). The above details are added in the *Discussion* section.

In addition, we agree that, for this pilot MS data, the type II errors could be large due to the small sample size (n=5). The above concern is added in the *Materials and Methods, Statistical analysis* section.

R1.3. I am curious as to why the authors chose to perform 1-tailed non-parametric tests rather than opt for the more conservative 2-tailed version?

In this study, we chose 1-tailed non-parametric testing as we specifically hypothesized a decrease in the strength of restrictions c in MS lesions as compared to NAWM, and an increase in the bulk diffusivity D_{∞} . Indeed, since (1) the inner axonal diameters in cross-sections with mitochondria present are larger than those without based on the previous histological study (Wang, et al., *IOVS*, 2003, doi: 10.1167/iovs.02-0333) and Figure 6c, and (2) mitochondria density is increased in MS lesions (Witte, et al., *Trend in Molecular Medicine*, 2013, doi:10.1016/j.molmed.2013.11.007), more variation in axon caliber is expected with a corresponding decrease in the strength of restrictions c . Similarly, we expect that the bulk diffusivity D_{∞} in MS lesions is larger than that in NAWM due to demyelination (Moll, et al., *Annals of Neurology*, 2011, doi: 10.1002/ana.22521).

Furthermore, we had to use a paired Wilcoxon-sign rank test with n=5 for the comparison of MS lesions and NAWM, as our dataset does not pass the normality test, and thus we cannot confidently use the paired t-test. For this specific non-parametric test with an n=5, the smallest possible P-value for a 1-tailed test is $0.5^5=0.03125$. We note, however, that for a conventional t-test, the P-value is more statistically significant (P-value=0.012 for one-tailed), but we decided we cannot confidently use the conventional t-test. Based on the discussion above, we now emphasize that the comparison of MS lesions and NAWM is exploratory due to the small sample size and report the smallest possible P-value in the *Materials and Methods, Statistical analysis* section.

Reviewer #2

I am not fully convinced by this conclusion, in spite of the great interest and beauty of the work.

MAJOR

R2.1 One of the problems is that the mouse CC provided estimates of local misalignment over a 20 μ m slice while the dMRI signal is extracted from a far thicker section which will certainly produce an estimation over-fitting on the Monte-Carlo simulations towards local inhomogeneities of the few extracted samples.

We thank the reviewer for raising this comment, and would like to clarify that we do not think there is overfitting, as we use in this study a very simple model with only two parameters, i.e. a linear fitting of the $1/\sqrt{t}$ power-law scaling to the simulated diffusivity time(t)-dependence. We observe this specific power-law scaling both through Monte-Carlo simulations and in in vivo diffusion MRI measurements. Rather, we interpret this comment as the concern that our small mouse brain EM sample (size: 36 μ m x 48 μ m x 20 μ m) is potentially *under-representative* as compared to the overall sample size that is probed with diffusion MRI.

To address this, we are currently in the process of performing Monte Carlo simulations inside realistic axonal shapes segmented from a 3d electron microscopy data of the human subcortical white matter. The human brain sample was obtained from the temporal gyrus of a patient during surgical operation for the removal of an epileptogenic focus in the hippocampus. The sample was trimmed to 2mm x 3mm x 0.2mm, scanned with serial scanning EM, and segmented using 2d/3d U-Net. This high quality human brain data of such a large length scale will answer the question raised by the reviewer, and the preliminary data submitted to the 2020 *ISMRM* annual meeting (please see the attached file “ISMRM_2020_abstract_278.pdf” and keep it confidential) have revealed the $1/\sqrt{t}$ power-law scaling to the simulated axial diffusivity time(t)-dependence, similar to this submission to the *Communications Biology*. This latest project is an independent project under different authorship and funding, that has been accepted as an oral presentation at the ISMRM conference. While we cannot include it in the current manuscript, we now refer to the abstract in the *Discussion* section.

R2.2 The second problem is that the mouse CC does not provide appropriate guidance for the work in humans since axon diameters are much more uniform in rodents than in primates. In conclusion, why is the radial diffusivity function much steeper for the midbody of CC than for other CC districts? Can this be due to some special structural feature of this sector of CC, in particular the greater variation in axon diameters, with very thick and thin axons interspersed? Indeed, the authors should demonstrate that the variation in longitudinal diffusivity are not due to changes in diameter across axons rather than along axons.

We thank the reviewer for this comment. While we agree that there could be differences between rodents and primates, we would like to emphasize that the $1/\sqrt{t}$ power-law scaling of the axial diffusivity seems a universal effect, observed in vivo in humans, as well as in simulations based on the human EM data mentioned in the previous comment R2.1 (please see the attached file), which is similar to the results and conclusions in this mouse brain study.

Furthermore, we would like to clarify that we do not show experimental data of the radial diffusivity (perpendicular to axons) in this study, but of the axial diffusivity time-dependence for the in vivo measurements in Figure 3, which show steeper $1/\sqrt{t}$ power-law scaling in the midbody of CC. This could indeed be due to the stronger caliber variation along individual axon in this brain region, as demonstrated in Figure 2 and explained in the *Discussion* section, and is an interesting observation that needs to be further understood, for which supporting 3d histology and analysis are needed.

In addition, we also would like to clarify that the diffusivity time-dependence along the fiber bundle is not due to the diameter distribution across axons. Indeed, highly aligned fibers with no caliber variations have no diffusivity time-dependence longitudinal to fibers even if the diameter distribution across fibers is very large, as illustrated here by Monte Carlo simulations in a micro-geometry composed of parallel straight cylinders (with no caliber variation along each cylinder) with a wide diameter distribution across cylinders (Figure R1a). The diameter distribution of parallel cylinders (Figure R1b) is the same as that of cross-sections of segmented axons in the main text. The simulation results in Figure R1c shows that the parallel straight cylinders have no diffusivity time-dependence along the fiber bundle, even with a very wide diameter distribution. This is because that, along the fiber bundle composed of parallel straight cylinders, diffusion along fibers is not hindered by any restrictions and is effectively Gaussian (at any time range), which is characterized by the lack of diffusivity time-dependence.

In contrast, the diffusion along the fiber bundle for realistic axonal shapes (Figure 2 in the main text) demonstrates significant diffusivity $1/\sqrt{t}$ -dependence, which is mainly due to the caliber variation of individual axon, and other confounding factors, such as undulation and orientation dispersion, have small effects on that, as shown in Figures 2 and 3. The above discussion is added in the *Discussion* section.

Figure R1. (a) Densely packed parallel straight cylinders (with no along-cylinder caliber variation for each individual cylinder). (b) Diameter distribution of straight cylinders in (a). This distribution is the same as that of cross-sections of segmented axons in the main text. (c) The simulated diffusivity along densely packed straight cylinders shows no time-dependence, a signature of Gaussian diffusion.

R2.3 Finally, a concern about the conclusions reached on synthetic data is that they cannot be directly generalized to human dMRI data, since the extra-axonal space may be a concomitant constituent of the time dependency measured in real data.

We agree with the reviewer that the observed diffusivity time-dependence of in vivo measurements could be contributed by not only intra-axonal signals but also extra-axonal ones. Although the extra-axonal signals are relatively smaller than the intra-axonal signals because of the shorter T_2 in extra-axonal space and long echo time applied in experiments (Veraart et al., Neuroimage 2018, doi: 10.1016/j.neuroimage.2017.09.030), the contribution of extra-axonal space may be non-negligible. Nonetheless, for the diffusivity time-dependence along the fiber bundle, we expect that the diffusivity time-dependence in the extra-axonal space is similar to that in intra-axonal space, since water molecules experience similar beading arrangement in either intra- or extra-axonal spaces. However, we cannot test it in realistic EM-based geometry, because the extra-axonal space of tissue samples shrinks during the tissue preparation for EM. To reasonably maintain the extra-axonal space, we are working with other groups for the novel tissue preparation pipeline, which still is an ongoing project. The above limitations are included in the *Discussion* section as the outlook of future directions.

MINOR

R2.4 What is c in eq 1 ? say it right away.

We thank the reviewer for the comment. The parameter c is the strength of restrictions. The above detail is added right after Equation 1.

Reviewer #3

The majority of my comments are related to how the manuscript is presented. Only the first comment below potentially bears on the soundness of the conclusions.

R3.1 Mitochondria were explicitly segmented in the 3D EM data. In order to incorporate the potential effects of mitochondria in simulations of water diffusion, they were modeled as structures with “fully permeable” membranes, but distinct from the cytoplasm in that the water T2 is 20 ms (rather than the T2 of 80 ms assumed for water in the cytoplasm). Work by Cerdan et al. is cited to support these choices of physical parameters. However, another characteristic of mitochondria described by the Cerdan group is its 15-30 times larger apparent viscosity than that of axoplasm, which leads to “reductions of more than one order of magnitude of the diffusion coefficient of water”. The authors do not seem to incorporate this difference between cytoplasmic and mitochondrial water in their simulations. This could be an important confound, given the correlation between mitochondrial location and regions of locally increased axon diameter.

We agree with the reviewer that the diffusivity difference between mitochondria and cytoplasm could be a potential confound, and took this opportunity to incorporate the effect of the mitochondria in a more realistic way. Now we assume that the diffusivity in mitochondria is 15 times smaller than that in cytoplasm (based on the viscosity differences in (Lopez, Mate and Cerden, J Bio Chem, 1996)), and we re-implemented our Monte Carlo simulations in CUDA C++ to account for the exchange between compartments with different diffusivities. The simulated diffusion metrics are slightly changed, but the conclusion is generally the same. The above details are added in *Results* and *Materials and Methods* sections.

R3.2 It would be helpful to see a figure that combines the experimental data with the data from simulations. Given the different behavior for different WM structures (Figure 4) and the different curves as parameters are varied in the simulations (Figures 2 and 3), a combined figure could help the reader appreciate how well the theory and experiment match, or how the WM structures differ from each other.

We thank the reviewer for the comment. We agree that combining experimental data with simulations is a comprehensive way of data visualization. On the other hand, we deemed that combining all Figures 2,3,4 into one big figure was a bit overwhelming and too big to fit on one page. Therefore, as a compromise of readability, Figures 2-3 for MC simulations in the last submission are edited as a combined Figure 2, and the original Figure 4 for in vivo MRI data is kept the same (now Figure 3). We will also pay attention to the final version having both figures to be placed close to each other in the article.

R3.3 The “pilot study” of Figure 6 does not contribute to this manuscript’s primary focus, to establish the role of variation in axon diameter in diffusion MRI measurements. I am in favor of removing this section, as it raises many distracting questions related to segmentation, the role of variation between WM structures, contributions of various other characteristics of WM pathology, etc.

We agree with the review that the clinical application of dMRI time-dependence measurements is not the primary focus of this study. However, the pilot MS data in Figure 5 (Figure 6 of last submission) serve as an example of in vivo human validation. As also mentioned in the response to R1.2, rather than aiming to propose another biomarker for MS lesions, we used MS data in this study to demonstrate that the strength of restrictions c (obtained from time-dependence diffusion MRI measurement) is significantly different between lesions and NAWM, and attribute this difference to the well-known *increase in mitochondria*, as a response to *demyelination* in MS lesions (Witte, et al., *Trend in Molecular Medicine*, 2013, doi:10.1016/j.molmed.2013.11.007). This observation may contribute to understanding the underlying pathological mechanisms taking place in MS lesion formation. In addition, our finding also suggests diffusion time-dependence measurements as potential biomarker suitable for monitoring other pathologies, such as traumatic brain injury and ischemic stroke, presenting increased neurite beadings due to other mechanisms (rather than mitochondrial increase). Therefore, we preferred to keep the section of MS data as its present form, but now also added the above details in the *Discussion*, and also emphasized the exploratory nature of this study including the limited sample size.

R3.4 In the results section devoted to fiber dispersion, a range of Watson distribution concentration parameters was explored. The concentration parameter obtained from the axons in the 3D EM data should be provided here for context.

We thank the reviewer for this comment, and have now added the concentration parameter based on the 3d EM data in the *Results* section.

R3.5 The last 4 paragraphs of the Introduction are unnecessary. A succinct summary might be helpful in the last paragraph of the Introduction, but detailed summaries such as the one provided is more common in engineering journals than they are (I think) in Communications Biology papers. Similarly, the 3rd paragraph of the Discussion is not needed. The Discussion section subheadings serve the purpose.

The last four paragraphs in Introduction are re-written into a succinct one. In addition, the third paragraph of the *Discussion* section is deleted.

R3.6 It might be helpful to provide 1-2 sentence transitions at the beginnings of paragraphs that review the different physical parameters (pages 5,6). As examples, the reason for evaluating the effect of fiber dispersion could be provided at the beginning of the paragraph that describes Figure 3. This and the next paragraph could provide a 1-sentence reminder of the information related to the parameter “c”. The next paragraph (beginning on page 6) abruptly begins a discussion of $K(t)$, which, the reader might want to be reminded, refers to the time dependence of kurtosis.

We agree with the reviewer that few sentence transitions improve the readability of paragraphs reviewing physical parameters. In the first subsection of the *Results* section, we now added a sentence to explain the reason for evaluating the effect of fiber dispersion and provide sentences to remind readers the meaning of time-dependence amplitude c in Equation 1. And in the beginning of the next paragraph, we added a sentence to remind readers that the kurtosis time-dependence is referred in this paragraph.

R3.7 On page 5, the text following the roman numeral II states that T2 contrast between axoplasm and mitochondria is removed by making $T2a = T2m$, but the numerical value is not provided. Presumably it is 80 ms, but the number should be given.

We thank the reviewer for noticing this oversight. For the micro-geometry II in Figure 2a, the T2 in cytoplasm and mitochondria are indeed both 80 ms. This information is now added on page 5.

R3.8 On pages 10-11, it was difficult to find where the meaning of the parameters $n_{i,j}$ and $x_{i,j}$ are defined.

The i_j is a common index notation of tensor components in physics. We agree with the reviewer that the definition of i_j could confuse readers since i alone is the index of axons. Therefore, we replace i_j with j_1, j_2, j_3 and j_4 , and provide definitions right after with appropriate references (Jensen et al., 2005, doi: 10.1002/mrm.20508; Jensen et al., 2010, doi:10.1002/nbm.1518).

Reviewers' comments:

Reviewer #1 (Remarks to the Author):

The revised manuscript adequately addresses the concerns I raised in my initial review.

Reviewer #2 (Remarks to the Author):

I am sorry to say that in spite of the efforts of the authors, the conclusions of the present study fail to respond to my main concern, that is that the effects on longitudinal, along axons diffusion may be due to diameter variations across axons rather than along individual axons. I acknowledge the efforts of responding to my R2.2 concern with Monte Carlo simulations on a model of parallel straight cylinders of varying diameters. Unfortunately, in real life axons of different diameters do not run parallel. Rather they undulate and cross each other. Therefore, the simulation provided does not eliminate my concern. Rather the authors should simulate diffusion along bundles of axons of different and constant diameter which also cross each other and undulate. This should be compared to diffusion along bundles of axons of different and varying diameters which also cross and undulate. I am conscious that such a simulation might show that diameter changes along and across axons cannot be separated with the current limitations on the resolution of dMRI, but this would be useful to know. I also notice that the regional variations in D_{∞} and c shown in Fig 4 are extremely noisy across subjects. The averaged antero-posterior differences most probably represent the higher spectrum of axon diameters in the midbody which contains axons originating from the motor and somatosensory cortex ranging between 0.4 and 5 μm according to Aboitjs et al (1992) rather than a higher change in axon diameters along individual axons in this callosal sector.

Reviewer #3 (Remarks to the Author):

The authors have addressed the comments of my review.

7 May 2020

Probing axon caliber variations and beading with time-dependent diffusion MRI

We sincerely thank reviewers #1 and #3 for accepting our submission, and reviewer #2 for the time, effort and suggestions for further improving the manuscript. We took suggestions into account and addressed all raised comments.

In particular, to investigate the effect of axonal diameter distribution on axial diffusivity time-dependence, we performed simulations of diffusion in undulating fibers (with or without caliber variations along individual fibers) and perfectly straight cylinders, with the consideration of diameter distribution across fibers and orientation dispersion (axial symmetric Watson distribution). The simulation results demonstrate that fibers with no caliber variations have negligible diffusivity time-dependence along fiber bundles, even for the highly dispersed case with very wide diameter distribution; in contrast, the axons with caliber variations show significant axial diffusivity time-dependence. Therefore, we conclude that the observed diffusivity time-dependence along axons is mainly contributed by caliber variations along individual axons, rather than diameter distribution across different axons.

As a reference, we have enclosed the manuscript in which the text additions supplied in response to the Reviewer critiques and suggestions are shown in **red**. Below, we copy each of the comments of the reviewer in *italic* font followed by our response in regular font.

Reviewer #2

R2.1 I am sorry to say that in spite of the efforts of the authors, the conclusions of the present study fail to respond to my main concern, that is that the effects on longitudinal, along axons diffusion may be due to diameter variations across axons rather than along individual axons. I acknowledge the efforts of responding to my R2.2 concern with Monte Carlo simulations on a model of parallel straight cylinders of varying diameters. Unfortunately, in real life axons of different diameters do not run parallel. Rather they undulate and cross each other. Therefore, the simulation provided does not eliminate my concern. Rather the authors should simulate diffusion along bundles of axons of different and constant diameter which also cross each other and undulate. This should be compared to diffusion along bundles of axons of different and varying diameters which also cross and undulate. I am conscious that such a simulation might show that diameter changes along and across axons cannot be separated with the current limitations on the resolution of dMRI, but this would be useful to know.

We thank the reviewer for reiterating this comment and agree that dispersion and undulation are a potential concern that can be addressed by comparing simulation results in orientationally dispersed fiber bundles composed of undulating fibers with or without caliber variations. Therefore, we performed additional simulations of orientationally dispersed fiber bundles composed of (1) realistic axonal shapes (geometry I in Figure 2) with undulations and caliber variations along each axon, (2) undulating fibers with no caliber variations along each fiber (geometry IV in Figure 2), and (3) perfectly straight cylinders with no undulations and caliber variations along individual fibers. The above three fiber bundle geometries have the same $2d$ cross-

sectional diameter distributions across fibers, and orientation distributions are based on the axially symmetric Watson distribution with dispersion angles $\theta = [0^\circ, 15^\circ, 30^\circ, 45^\circ]$.

Simulation results (Figure R1) shows for fiber bundles composed of fibers with only axonal undulations or perfectly straight cylinders very small diffusivity changes along the main direction of fiber bundles at clinical diffusion times $t = 20\text{-}100$ ms ($\sim 0.5\%$ diffusivity change for $\theta = 45^\circ$), whereas for the fiber bundle of realistic IAS simulation results demonstrate significant diffusivity time-dependence along axons within the same time range ($\sim 5\%$ diffusivity change for $\theta = 45^\circ$). Therefore, we conclude that the diffusivity time-dependence along axons is mainly contributed by the caliber variations along individual axons, rather than the diameter distribution across axons. The above conclusion is added in *Discussion* section, and the details and extra figure are added in *Supplementary Information*. Also, we updated the generation algorithm of undulating only fibers for better simulation precision.

Figure R1: The diffusivity time-dependence along axons is dominated by caliber variations along individual axons, rather than diameter distribution across different axons, demonstrated here by Monte Carlo simulations in fiber bundles composed of (1) realistic IAS with caliber variations and axonal undulations (geometry I in Figure 2a), (2) fibers with only axonal undulations (geometry IV in Figure 2a), and (3) perfectly straight cylinders. The above three fiber bundles have the same $2d$ cross-sectional diameter distribution and orientation dispersion (Watson distribution). Simulation results show that fiber bundles with no caliber variations along individual fibers have negligible diffusivity time-dependence at clinical diffusion time $t = 20\text{-}100$ ms, whereas realistic IAS has significant diffusivity $1/\sqrt{t}$ -dependence along axons within the same time range.

R2.2 I also notice that the regional variations in D_{∞} and c shown in Fig 4 are extremely noisy across subjects. The averaged antero-posterior differences most probably represent the higher spectrum of axon diameters in the midbody which contains axons originating from the motor and somatosensory cortex ranging between 0.4 and $5 \mu\text{m}$ according to Aboitiz et al (1992) rather than a higher change in axon diameters along individual axons in this callosal sector.

Admittedly, the regional variations in D_{∞} (bulk diffusivity at long times) and c (strength of restrictions) are noisy for individual subjects; however, we are still able to observe the general trends across the corpus callosum (CC) for the average over all subjects, which relates remarkably well to the pattern of axonal density in CC observed in histology (Aboitiz, et al., 1992, ref 34), of axonal volume fraction in CC estimated via dMRI (Barazany, et al., 2009, ref 18; Fieremans, et al., 2011, ref. 35), as well as the higher spectrum of large axon diameters in the midbody according

to Aboitz et al (1992). On the one hand, the high-low-high trend in D_∞ in CC could be related to the pattern of axonal density in CC observed in histology assuming the axial diffusivity in IAS is larger than that in EAS (Veraart, et al., 2018, ref 28). On the other hand, the low-high-low trend in c in CC could also be related with the bead width and/or distance between local caliber maxima along individual axons, which unfortunately cannot be supported or rejected by 2-dimensional histology (Aboitz et al., 1992, ref 34). For example, fiber bundles composed of (1) caliber-varying axons or (2) perfectly straight cylinders can have exactly the same 2-dimensional cross-sectional diameter distribution. Three-dimensional histology and analysis in different regions of CC are needed in the future to better understand our empirical observation of the trends across the corpus callosum. The above details are added and highlighted in *Discussion* section.

REVIEWERS' COMMENTS:

Reviewer #2 (Remarks to the Author):

The authors have convincingly responded to my main comments concerning the effects of individual axonal diameter changes vs axonal misalignment. The effects on the human A-P CC trends remain incompletely answered and I agree that 3D histological information on caliber of individual axons on different CC sectors is required

Point-by-point response to comments / list of changes

27 May 2020

A time-dependent diffusion MRI signature of axon caliber variations and beading

We sincerely thank reviewer #2 for additional feedback and update the manuscript to take this into account. As a reference, we have enclosed the manuscript in which the text additions supplied in response to the Reviewer critiques and suggestions are shown in red.

Reviewer #2

R2.1 The authors have convincingly responded to my main comments concerning the effects of individual axonal diameter changes vs axonal misalignment. The effects on the human A-P CC trends remain incompletely answered and I agree that 3D histological information on caliber of individual axons on different CC sectors is required.

We agree with the reviewer that the trends of time-dependent parameters across human corpus callosum remain incompletely answered, and the 3D histology study is required to answer this question in the future. The above details are added and highlighted in the *Discussion* section.